# Single-cell 'omic profiles of human aortic endothelial cells in vitro and human atherosclerotic lesions ex vivo reveal heterogeneity of endothelial subtype and response to activating perturbations

**Maria L Adelus[1,2], Jiacheng Ding[1], Binh T Tran[1], Austin C Conklin[1], Anna K Golebiewski[1], Lindsey K Stolze[1], Michael B Whalen[1], Darren A Cusanovich[1,3], Casey E Romanoski[1,2,3]\***

[1]The Department of Cellular and Molecular Medicine, The University of Arizona, Tucson, United States; [2]The Clinical Translational Sciences Graduate Program, The University of Arizona, Tucson, United States; [3]Asthma and Airway Disease Research Center, The University of Arizona, Tucson, United States

**\*For correspondence:** cromanoski@arizona.edu

**Competing interest:** The authors declare that no competing interests exist.

**Preprint posted** 10 August 2023 **Sent for Review** 22 August 2023 **Reviewed preprint posted** 04 December 2023 **Reviewed preprint revised** 04 March 2024 **Version of Record published** 05 April 2024

**Abstract** Heterogeneity in endothelial cell (EC) sub-phenotypes is becoming increasingly appreciated in atherosclerosis progression. Still, studies quantifying EC heterogeneity across whole transcriptomes and epigenomes in both in vitro and in vivo models are lacking. Multiomic profiling concurrently measuring transcriptomes and accessible chromatin in the same single cells was performed on six distinct primary cultures of human aortic ECs (HAECs) exposed to activating environments characteristic of the atherosclerotic microenvironment in vitro. Meta-analysis of single-cell transcriptomes across 17 human ex vivo arterial specimens was performed and two computational approaches quantitatively evaluated the similarity in molecular profiles between heterogeneous in vitro and ex vivo cell profiles. HAEC cultures were reproducibly populated by four major clusters with distinct pathway enrichment profiles and modest heterogeneous responses: EC1-angiogenic, EC2-proliferative, EC3-activated/mesenchymal-like, and EC4-mesenchymal. Quantitative comparisons between in vitro and ex vivo transcriptomes confirmed EC1 and EC2 as most canonically EC-like, and EC4 as most mesenchymal with minimal effects elicited by siERG and IL1B. Lastly, accessible chromatin regions unique to EC2 and EC4 were most enriched for coronary artery disease (CAD)-associated single-nucleotide polymorphisms from Genome Wide Association Studies (GWAS), suggesting that these cell phenotypes harbor CAD-modulating mechanisms. Primary EC cultures contain markedly heterogeneous cell subtypes defined by their molecular profiles. Surprisingly, the perturbations used here only modestly shifted cells between subpopulations, suggesting relatively stable molecular phenotypes in culture. Identifying consistently heterogeneous EC subpopulations between in vitro and ex vivo models should pave the way for improving in vitro systems while enabling the mechanisms governing heterogeneous cell state decisions.

## eLife assessment

This is a **fundamental** resource of snRNA-seq and chromatin accessibility data from human aortic endothelial cells (ECs), treated with relevant perturbations such as IL1b, TGFB2, or siERG. The authors show that ECs can be categorized by distinct subpopulations of differing plasticity. The support for the existence of these subpopulations is **compelling**, supported also by three publicly

available scRNA-seq datasets, and differential enrichment of coronary artery disease associated SNPs in open chromatin in these subpopulations.

## Introduction

Endothelial cells (ECs) in the vascular endothelium maintain hemostasis, mediate vasodilation, and regulate the migration of leukocytes into tissues during inflammation. Dysfunctions of the endothelium are a hallmark of the aging process and are also an important feature of diseases, including atherosclerosis. Atherosclerosis is an inflammatory process fueled by cholesterol and leukocyte accumulation in the sub-endothelial layer of arteries. It is the underlying pathobiology of ischemic heart disease and the leading cause of morbidity and mortality worldwide due to heart attack and stroke (*Brown et al., 2020*; *Hajra et al., 2000*; *Birdsey et al., 2015*). Atherosclerosis of the coronary arteries is estimated to be about 50% genetic, with hundreds of genomic loci contributing to genetic risk (*Marenberg et al., 1994*; *Aragam et al., 2022*; *Tcheandjieu et al., 2022*). A major opportunity for better understanding the molecular basis for how disease progresses lies in identifying the genomic and downstream functions impaired by risk variants in disease-relevant cell types. Genetic studies are increasingly suggesting that a significant proportion of genetic risk for atherosclerosis is encoded in perturbed functions of vascular ECs (*Aragam et al., 2022*; *Tcheandjieu et al., 2022*; *Kessler and Schunkert, 2021*).

Single-cell sequencing technologies have begun to characterize the extent of EC molecular diversity in vitro and in vivo (*Zhao et al., 2018*; *Li et al., 2019*; *Kalluri et al., 2019*; *Liu et al., 2021*; *Kalucka et al., 2020*; *Rohlenova et al., 2020*; *Zhao et al., 2021a*; *Xu et al., 2020*; *Cheng et al., 2021*; *Khan et al., 2019*; *Andueza et al., 2020*; *Tombor et al., 2020*). Genetically engineered, lineage-traced mouse models have also been instrumental for identifying which cells in atherosclerotic plaques arose from EC origin. These studies have demonstrated that many cells of EC origin in plaques lack canonical EC marker genes and luminal location (*Evrard et al., 2016*; *Chen et al., 2012*). As many as one-third of mesenchymal-like cells in plaques have been reported to be of endothelial origin (*Evrard et al., 2016*), suggesting that phenotypic transition from endothelial to mesenchymal (EndMT) is a feature of atherosclerosis; however, whether EndMT is a cause or bystander of atherogenesis or plaque rupture is not fully understood. Although lineage tracing is not possible in humans, immunocytochemical techniques suggest that EC heterogeneity is prevalent in atherosclerotic vessels. These studies have described an unexpectedly large number of cells co-expressing pairs of endothelial and mesenchymal proteins, including fibroblast-activating protein/von Willebrand factor (FAP/VWF), fibroblast-specific protein-1/VWF (FSP-1/VWF), FAP/platelet-endothelial cell adhesion molecule-1 (CD31 or PECAM-1), FSP-1/CD31 (*Evrard et al., 2016*), phosphorylation of TGFB signaling intermediary SMAD2/FGF receptor 1 (p-SMAD2/FGFR1) (*Chen et al., 2015*), and α-smooth muscle actin (αSMA)/PECAM-1 (*Moonen et al., 2015*). An important implication of this result is that the use of canonical EC markers to isolate or identify ECs will likely omit certain EC populations. The extent of EC molecular and functional heterogeneity within a tissue during homeostasis and during disease is not well understood. One notable study exemplifying EC heterogeneity demonstrated that the EC-marker gene von Willebrand Factor (*VWF*) was expressed only in a subset of ECs from the same murine vessel, and the penetrance of *VWF* expression across ECs was tissue-specific (*Yuan et al., 2016*). In a related study, expression of the leukocyte adhesion molecule *VCAM-1* was found to be upregulated by the pro-inflammatory cytokine tumor necrosis factor α only in some of the ECs of a monolayer (*Turgeon et al., 2020*). In both studies, variability in DNA methylation on CpG dinucleotides at the gene promoters negatively correlated with *VWF* and *VCAM-1* expression. These findings raise the question as to how many molecular programs exist within ECs of a same tissue or culture, how this heterogeneity influences response to cellular perturbations, and what factors regulate these cellular states.

There are notable benefits and limitations for studying heterogeneity using in vitro and in vivo approaches in atherosclerosis research. In vitro approaches provide unique opportunities for interrogating consequences of genetic and chemical perturbations in highly controlled environments and are adept at identifying mechanistic relationships on accelerated timelines. In vivo approaches benefit from the complexity of the crosstalk among all cell types and tissues of the organism and are adept for identifying how perturbations manifest in living systems. It reasons that the integration of results from both approaches will best accelerate discovery. However, comprehensive analysis comparing

heterogeneity of vascular ECs observed in vivo and in vitro remains unexplored. In the current study, we performed meta-analysis on four human in/ex vivo single-cell transcriptomic datasets (*Pan et al., 2020*; *Alsaigh et al., 2022*; *Chowdhury et al., 2022*; *Wirka et al., 2019*), containing 17 arterial samples, from mild-to-moderate calcified atherosclerotic plaques to evaluate the ability of the in vitro EC models to recapitulate molecular signatures observed in human atherosclerosis.

Human aortic endothelial cells (HAECs) are among the most appropriate cell type for in vitro modeling of the arterial endothelium in atherosclerosis research insofar as they are human cells, they are more readily available than coronary artery ECs, they are not of venous origin like human umbilical vein ECs, and they can be isolated from explants of healthy donor hearts during transplantation. We set forth in the current study to quantify heterogeneity among HAECs using multimodal sequencing that simultaneously measures transcripts using RNA-seq and accessible chromatin using ATAC-seq from the same barcoded nuclei. To provide estimates for heterogeneity due to genetic background, we molecularly phenotyped HAECs from six genetically distinct human donors. We also quantified single-cell responses to three perturbations known to be important in EC biology and atherosclerosis. The first was activation of transforming growth factor beta (TGFB) signaling, which is a hallmark of phenotypic transition and a regulator of EC heterogeneity (*Evrard et al., 2016*; *van Meeteren and ten Dijke, 2012*). The second was stimulation with the pro-inflammatory cytokine interleukin-1 beta (IL1B), which has been shown to model inflammation and EndMT in vitro (*Bujak et al., 2008*; *Bujak and Frangogiannis, 2009*; *Maleszewska et al., 2013*; *Chaudhuri et al., 2007*; *Sánchez-Duffhues et al., 2019*), and whose inhibition reduced adverse cardiovascular events in a large clinical trial (*Ridker et al., 2017*). The third perturbation utilized in our study was knockdown of the ETS-related gene (*ERG*), which encodes a transcription factor of critical importance for EC fate specification and homeostasis (*Sperone et al., 2011*; *Fish et al., 2017*; *Lathen et al., 2014*; *Vijayaraj et al., 2012*; *Hogan et al., 2017*).

Lastly, we examine whether epigenetic landscapes among heterogeneous EC subtypes observed in this study were differentially enriched for coronary artery disease (CAD) genetic risk variants. Taken together, this study provides evidence that EC heterogeneity is prevalent in vivo and in vitro and that not all ECs respond similarly to activating perturbations.

## Results

### EC single-cell transcriptomic profiles reveal a heterogeneous population

To systematically uncover the heterogeneity of molecular landscapes in ECs at single-cell resolution, we cultured primary HAECs isolated from luminal digests of ascending aortas from six de-identified heart transplant donors at low passage (passages 3–6) (*Navab et al., 1988*; *Figure 1A*). Using the 10X Genomics multiome kit (*Genomics x, 2022a*), single-nucleus mRNA expression (snRNA-seq) and chromatin accessibility (snATAC-seq) data were collected simultaneously for a total of 15,220 nuclei after stringent quality control ('Materials and methods'). RNA and ATAC data were integrated separately by treatment condition and then with each other as reported previously ('Materials and methods'; *Hao et al., 2021*).

snRNA-seq libraries were sequenced to a median depth of 29,732–84,476 reads and 2481–3938 transcripts per nucleus (*Supplementary file 1a and b*). Five distinct EC subtypes (EC1, EC2, EC3, EC4, and EC5) were detected from the fully integrated dataset, which included all donors, treatments, and data types (*Figure 1B*). Subtypes EC1 and EC3 comprised cells from all donors, whereas EC2 and EC4 contained cells from most donors, and EC5 was nearly exclusively populated by cells from a single donor (*Figure 1C*, *Supplementary file 1c*). Because we do not observe EC5 across multiple individuals, we chose not to focus additional analysis on this subtype. Pathway enrichment of marker genes revealed EC1 to exhibit an angiogenic phenotype (WP4331, p-value $4.0 \times 10^{-9}$; GO:0038084, p-value $1.5 \times 10^{-9}$) with enriched transcripts including *KDR*, *GAB1*, *PGF*, and *NRP2* (*Figure 1D–G*, *Figure 1—figure supplement 1A*). EC2 was enriched in proliferation (GO:1903047, p-value $7.4 \times 10^{-35}$) with characteristic markers *CENPE*, *CENPF*, *KIF11*, *KIF4A,* and *TOP2A* (*Figure 1D–G*, *Figure 1—figure supplement 1A*). EC3 displayed enrichment in the 'regulation of smooth muscle cell proliferation' (GO:0048660; p-value $1.1 \times 10^{-10}$) (*Figure 1F*). From the top 200 differentially expressed genes (DEGs) for EC3, we observed additional pathways enriched, including NABA CORE MATRISOME

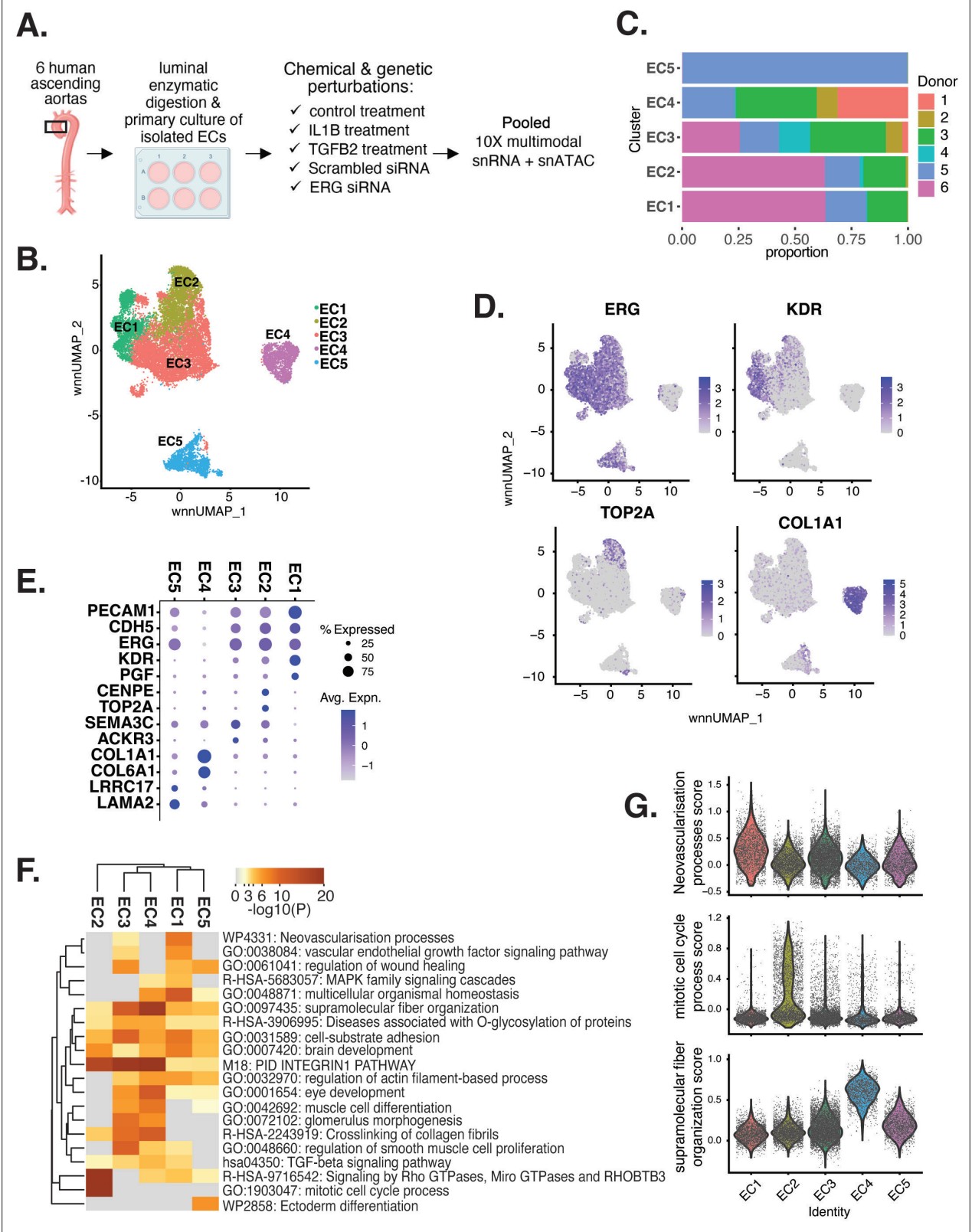

**Figure 1.** Human aortic endothelial cell (HAEC) transcriptomic profiling discover a heterogeneous cell population. (**A**) Schematic diagram of the experimental design. Endothelial cells (ECs) were isolated from six human heart transplant donor's ascending aortic trimmings and treated with IL1B, TGFB2, or siERG (ERG siRNA) for 7 d. (**B**) Weighted nearest-neighbor UMAP (wnnUMAP) of aggregate cells from all perturbations and donors is shown. Each dot represents a cell, and the proximity between each cell approximates their similarity of both transcriptional and epigenetic profiles. Colors

*Figure 1 continued on next page*

*Figure 1 continued*

denote cluster membership. (**C**) The proportion of cells from each donor for each EC subtype. (**D**) Gene expression across top markers for each cluster including pan EC (*ERG*), EC1 (*KDR*), EC2 (*TOP2A*), and EC4 (*COL1A1*). (**E**) Top markers for pan EC (*PECAM1*, *CDH5*, *ERG*), EC1 (*KDR*, *PGF*), EC2 (*CENPE*, *TOP2A*), EC3 (*SEMA3C*, *ACKR3*), EC4 (*COL1A1*, *COL6A1*), and EC5 (*LRRC17*, *LAMA2*). The size of the dot represents the percentage of cells within each EC subtype that express the given gene, while the shade of the dot represents the level of average expression ('Avg. Expn.' in the legend). (**F**) Heatmap of pathway enrichment analysis (PEA) results from submitting top 200 differentially expressed genes (DEGs; by ascending p-value) between EC subtypes. Rows (pathways) and columns (EC subtypes) are clustered based on -Log$_{10}$(P). (**G**) Violin plots of top Metascape pathway module scores across EC subtypes. Module scores are generated for each cell barcode with the Seurat function AddModuleScore().

The online version of this article includes the following figure supplement(s) for figure 1:

**Figure supplement 1.** Marker genes and pathways for endothelial clusters.

**Figure supplement 2.** Violin plot of *XIST* showing expected expression in female in vitro donor cells (1 and 3) and lack of expression in male in vitro donor cells (2, 4, 5, and 6).

(M5884; p-value $1 \times 10^{-34}$) and locomotion (GO:0040011; p-value $1.2 \times 10^{-15}$), suggesting an activated mesenchymal-like phenotype (*Figure 1—figure supplement 1B and C*). A fourth subset, EC4, demonstrates enrichment in ECM organization (GO:0097435; p-value $3.2 \times 10^{-19}$), a process characteristic of mesenchymal cells, with distinctive expression of collagen genes, including *COL1A1*, *COL1A2*, *COL3A1*, and *COL5A1* (*Figure 1D–G*, *Figure 1—figure supplement 1A*; *Dahal et al., 2017*; *Kovacic et al., 2019*). Top marker genes and pathways for each EC subtype are in *Supplementary file 1d and e*. These observations are in line with previous reports of angiogenic, proliferative, mesenchymal, and pro-coagulatory EC subtypes within ex vivo models (*Li et al., 2019*; *Kalluri et al., 2019*; *Zhao et al., 2021a*; *Tombor et al., 2020*; *Bondareva et al., 2022*) and underscore the heterogeneity of transcriptomic profiles in cultured HAECs.

## EC subtypes exhibit distinct open chromatin profiles and enriched motifs

To investigate how different transcriptional signatures across ECs correspond to distinct chromatin states, we utilized the snATAC-seq portion of the multiome dataset. The snATAC-seq data were sequenced to a median depth of 22,939–126,122 reads with 3480–19,259 peaks called per nucleus (*Supplementary file 1b and f*). Of the 204,904 total identified peaks, 13,731 were differential across subtypes, with 79–8091 peaks uniquely accessible per EC subtype (*Supplementary file 1h*). Over 80% of total peaks were intergenic or intronic (*Figure 2A and B*) and most unique peaks were from EC2 and EC4.

Transcription factor (TF) motif enrichment analysis using Signac (*Stuart et al., 2021*) was performed on differentially accessible regions (DARs) per EC subtype (*Figure 2C*). It is important to note that TFs within a TF family may share DNA-binding motifs and may not be distinguished by motifs alone. As a result, TF names from the Jaspar database (*Fornes et al., 2020*) indicate the TF family. We find the basic helix-loop-helix (*bHLH*) motif defined by the core sequence CANNTG enriched in EC1 peaks, including enrichments for ASCL2 (adjusted p-value $3.9 \times 10^{-50}$), TCF12 (adjusted p-value $1.7 \times 10^{-21}$), and BHLHE22(var.2) (adjusted p-value $5.7 \times 10^{-48}$) (*Figure 2C and D*). ETS motifs, including ETV1 (adjusted p-value $3.2 \times 10^{-42}$ and $5.3 \times 10^{-249}$, for EC1-2, respectively), SPIB (adjusted p-value $7.9 \times 10^{-22}$ and $2.5 \times 10^{-236}$, respectively), and GABPA (adjusted p-value $2.7 \times 10^{-41}$ and $4.3 \times 10^{-244}$, respectively), were also enriched in EC1 as well as in EC2 peaks. These data are consistent with known roles for ETS TFs, including ERG and FLI1, in governing angiogenic and homeostatic endothelial phenotypes (*Nagai et al., 2018*). Given that *ERG* expression (*Figure 1E*) correlated with incidence of the ETS motif in open chromatin (*Figure 2D*) across the nuclei, ERG is likely driving the EC1-2 sub-phenotypes. The near-exact match in motifs between the ETV1 motif position weight matrix in Jaspar and the de novo enriched motif from ERG ChIP-seq in human aortic ECs (*Hogan et al., 2017*) further supports this conclusion (*Figure 2E*). In addition to ETS motifs, EC2 was enriched in ZFX (adjusted p-value $4.2 \times 10^{-86}$) and ZNF148 (adjusted p-value $1.1 \times 10^{-126}$), which are C2H2 zinc finger motifs. C2H2 zinc finger motifs, as well as KLF4 (adjusted p-value $5.4 \times 10^{-32}$ and $8.4 \times 10^{-135}$, for EC1-2, respectively), also show enrichment in EC1 and EC2. EC3 peaks are enriched for GATA motifs including GATA4 (adjusted p-value $3.1 \times 10^{-8}$), GATA5 (adjusted p-value $8 \times 10^{-11}$), GATA1::TAL1 (adjusted p-value $1.8 \times 10^{-6}$), and bHLH motif BHLHE22(var.2) (adjusted p-value 0.01). EC4 open regions were uniquely enriched for TEA domain (TEAD) factors comprised of motifs named TEAD2 (adjusted p-value $1.2 \times 10^{-238}$),

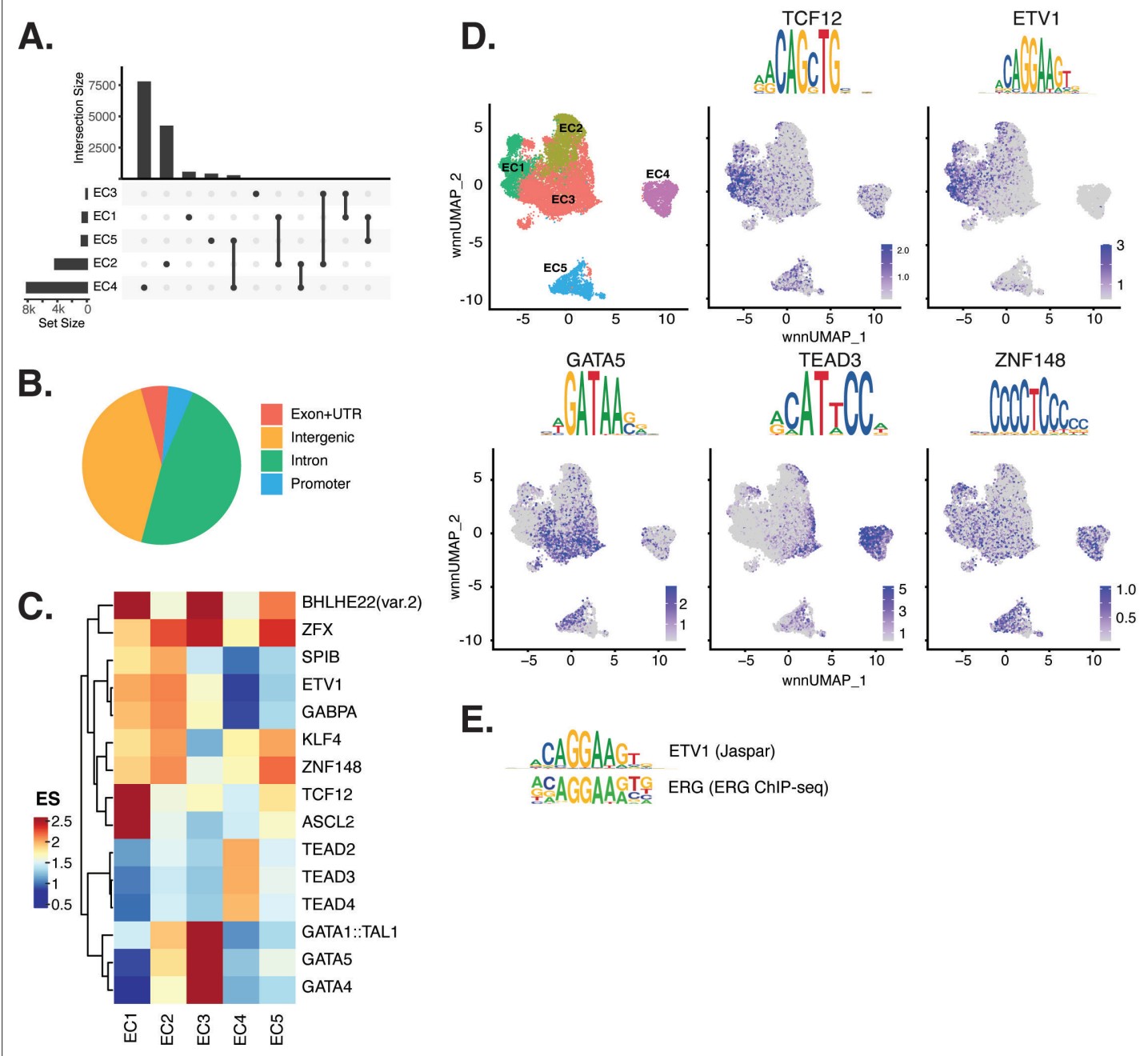

**Figure 2.** Endothelial cells (ECs) have epigenetically distinct cell states. (**A**) Upset plot of differential peaks across EC subtypes. Intersection size represents the number of genes at each intersection, while set size represents the number of genes for each EC subtype. (**B**) Genomic annotation for the complete peak set. (**C**) Heatmap of top transcription factors (TFs) from motif enrichment analysis for marker peaks in each EC subtype. Top TFs for each EC subtype are selected based on ascending p-value. Rows (TFs) and columns (EC subtype) are clustered based on enrichment score (ES). (**D**) Feature plots and position weight matrices (PWMs) for top TF binding motifs for EC1 (TCF12), EC2 (ETV1), EC3 (GATA5), and EC4 (TEAD3). Per-cell motif activity scores are computed with chromVAR, and motif activities per cell are visualized using the Signac function FeaturePlot. (**E**), PWMs comparing Jaspar 2020 ETV1 motif to ERG motif reported in Hogan et al.

TEAD3 (adjusted p-value $2.1 \times 10^{-306}$), and TEAD4 (adjusted p-value $6.9 \times 10^{-252}$) (*Figure 2C and D*). Notably, TEAD factors have been found as enriched in vascular smooth muscle cells (VSMCs) (*Wirka et al., 2019*; *Örd et al., 2021*), which is consistent with EC4 having the most mesenchymal phenotype of our EC subtypes.

Taken together, these data demonstrate that EC1 and EC2 are the subtypes most canonically like 'healthy' or angiogenic ECs insofar as they exhibit ETS motif enrichments. Additionally, we conclude that EC4 is the most mesenchymal EC insofar as it exhibits TEAD factor enrichments.

## EC-activating perturbations modestly shift cells into the EC3 subtype

Embedded in the dataset of this study were three experimental conditions known to promote EndMT along with their respective controls. Each experimental condition was administered to between three and five genetically distinct HAEC cultures. The conditions included 7-day exposure to IL1B (10 ng/ml), 7-day exposure to TGFB2 (10 ng/ml), and 7-day siRNA-mediated knockdown of ERG (siERG). The control for IL1B and TGFB2 treatments was 7-day growth in matched media lacking cytokine and the control for the siERG condition was transfection with scrambled RNA.

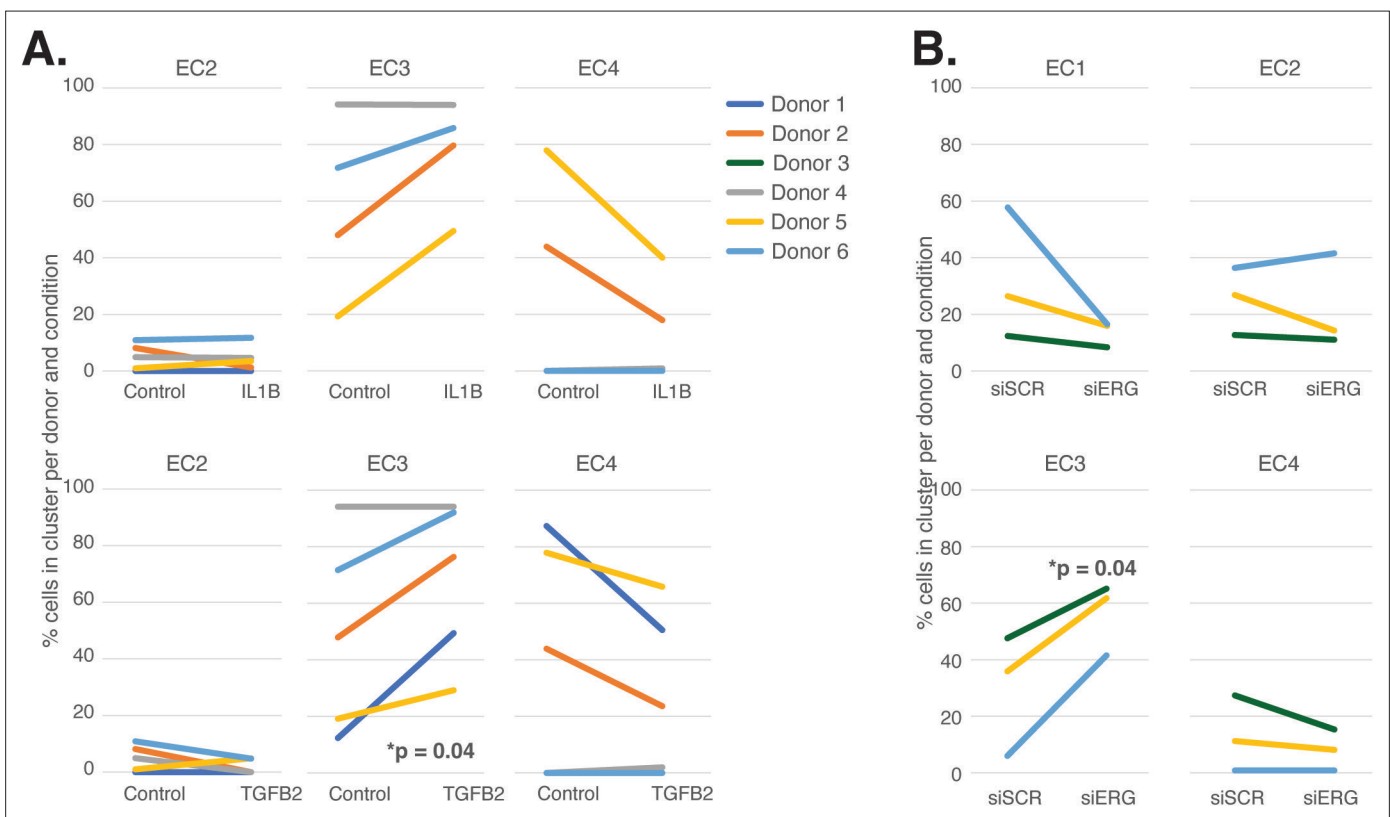

**Figure 3.** Endothelial cell (EC)-activating perturbations modestly shift cells into the EC3 subtype. (**A**) The proportion of cells in 7-day control and 7-day IL1B treatment are shown per human aortic endothelial cell (HAEC) donor and cluster on the top and for 7-day control and 7-day TGFB2 on the bottom. (**B**) The proportion of cells in 7-day siSCR control and 7-day siERG knockdown are shown per HAEC donor and cluster. EC1 was omitted in (**A**) due to lack of cells in both conditions.

The online version of this article includes the following source data and figure supplement(s) for figure 3:

**Source data 1.** Annotated western blots for **Figure 3—figure supplement 3B** where the leftmost six wells are shown from left to right as (1) the protein ladder (labeled in kD), (2) the lipofectamine transfected control, (3) the scrambled siRNA control, (4, 5) two lanes using different siRNAs against the TCF4 gene (not relevant to these studies), and lastly, (6) siRNA against ERG.

**Source data 2.** Original western blots for ERG, annotated and uncropped.

**Source data 3.** Original western blots for H3, annotated and uncropped.

**Source data 4.** Original western blots for ERG, unannotated and uncropped.

**Source data 5.** Original western blots for H3, unannotated and uncropped.

**Figure supplement 1.** Principal component analysis using RNA-seq data.

**Figure supplement 2.** Receptor expression profiles across endothelial clusters.

**Figure supplement 3.** Validation of ERG knockdown.

The UMAP presented in *Figure 1* includes all the nuclei profiled across donors and conditions. We hypothesized that EC4, the most mesenchymal cluster, would be enriched for cells exposed to IL1B, TGFB2, and/or siERG relative to the controls thereby consistent with the hypothesis that the EC4 subtype were a consequence of EndMT. Detailed in *Figure 3A and B* are the relative proportions of cells from each experimental condition and donor by cluster. Contrary to our hypothesis, the EC4 cluster was not enriched for cells that were treated with cytokine or siERG relative to the controls; in fact, there is a nonstatistically significant trend for decreased numbers of EC4 cells from these conditions relative to controls insofar as all the donors with cells in EC4 show diminished proportions upon perturbation (*Figure 3*). The one cluster exhibiting increased proportions of cells upon perturbations was EC3, with three of four EC IL1B-exposed donors having increased proportions in EC3 (p=0.08 by two-sided paired *t*-test; *Figure 3A*), four of five TGFB2-exposed donors having increased proportions (p=0.04 by two-sided paired *t*-test; *Figure 3A*), and three of three donors having increased EC3 proportions upon ERG knockdown (*Figure 3B*).

In addition to heterogeneity across EC clusters, data in *Figure 3* underscores that there is heterogeneity among EC cultures. To quantify this effect, we performed principal component analysis (PCA) to evaluate the overall contributions that donor and experimental conditions have on variance in this dataset. We found that pro-EndMT perturbations elicited greater variance in RNA expression (38–56% of variance) than donor (17%–27% variance) (*Figure 3—figure supplement 1A–C*), supporting that the transcriptional and epigenetic programs elicited by experimental conditions have a greater overall consequence than donor. This finding provides the opportunity to elucidate how different EC clusters respond to pro-EndMT exposures across genetically distinct ECs.

## Pro-EndMT perturbations in vitro elicit EC subtype-specific transcriptional responses

We next sought to evaluate the similarities and differences among pro-EndMT perturbations and evaluate the transcriptional response elicited in each EC subtype. Differential gene expression analysis was performed using pseudo-bulked profiles grouped by donor, subcluster, and experimental groupings (*Supplementary file 1i*).

Overall, we found heterogeneity in transcriptional responses across EC subtypes. While EC1 and EC2 transcripts were predominantly perturbed by siERG, the greatest number of transcripts differentially expressed in EC3 were those responsive to IL1B, though siERG and TGFB2 also regulated tens to hundreds of transcripts in EC3. In contrast, transcripts in EC4 were predominantly responsive to TGFB2 (*Figure 4A*, *Supplementary file 1i*). With respect to EC4, we questioned whether transcripts were predominantly responsive to TGFB2 due to differences in expression of TGFB receptors. While we observed increased TGFBR1 expression in EC4, we observed relatively less expression of TGFBR2 and ACVRL1 in EC4 when compared to EC1, EC2, and EC3 (*Figure 3—figure supplement 2A*). We next questioned whether EC3 transcripts were predominantly responsive to IL1B due to differences in IL1B receptor expression. Notably, we did not observe differences in IL1B receptor expression, suggesting that their transcription is not responsible for divergent EC responses across EC subtypes (*Figure 3—figure supplement 2B*). Interestingly, we did observe differential expression of IL1RL1 in EC2, which may influence EC2 response to cytokine (*Figure 3—figure supplement 2B*).

When comparing enriched pathways across perturbations, we observed that over 80% of transcripts differentially expressed by a treatment in EC4 were in response to TGFB2 (*Figure 4A*, *Supplementary file 1i*). TGFB2-affected transcripts for EC4 were enriched in invadopodia formation (R-HAS-8941237; p-value $2.7 \times 10^{-7}$) and anchoring fibril formation (R-HAS-2214320; p-value $3.6 \times 10^{-7}$) (*Figure 4B*). Notably, TGFB2-affected genes for EC3 share several mesenchymal-related enriched pathways with TGFB2-affected genes for EC4, including actin cytoskeleton organization (GO:0030036; p-value $4.4 \times 10^{-7}$), NABA CORE MATRISOME (M5884; p-value $2.8 \times 10^{-7}$), and ECM organization (R-HSA-1474244; p-value $5.4 \times 10^{-7}$). TGFB2-attenuated transcripts unique to EC3 were enriched in platelet activation (GO:0030168; p-value $1.4 \times 10^{-4}$) (*Figure 4B*).

Most transcripts affected in EC3 were responsive to IL1B (*Figure 4A*). Importantly, several EC3 genes differentially expressed with IL1B were also affected with siERG (*Figure 4A*). IL1B-affected transcripts in EC3 are not enriched in mesenchymal-like pathways (*Figure 4C*). However, EC3 IL1B-attenuated genes are enriched in blood vessel development (GO:0032502; p-value $5.1 \times 10^{-11}$), indicating that this perturbation still has anti-endothelial effects (*Figure 4C*).

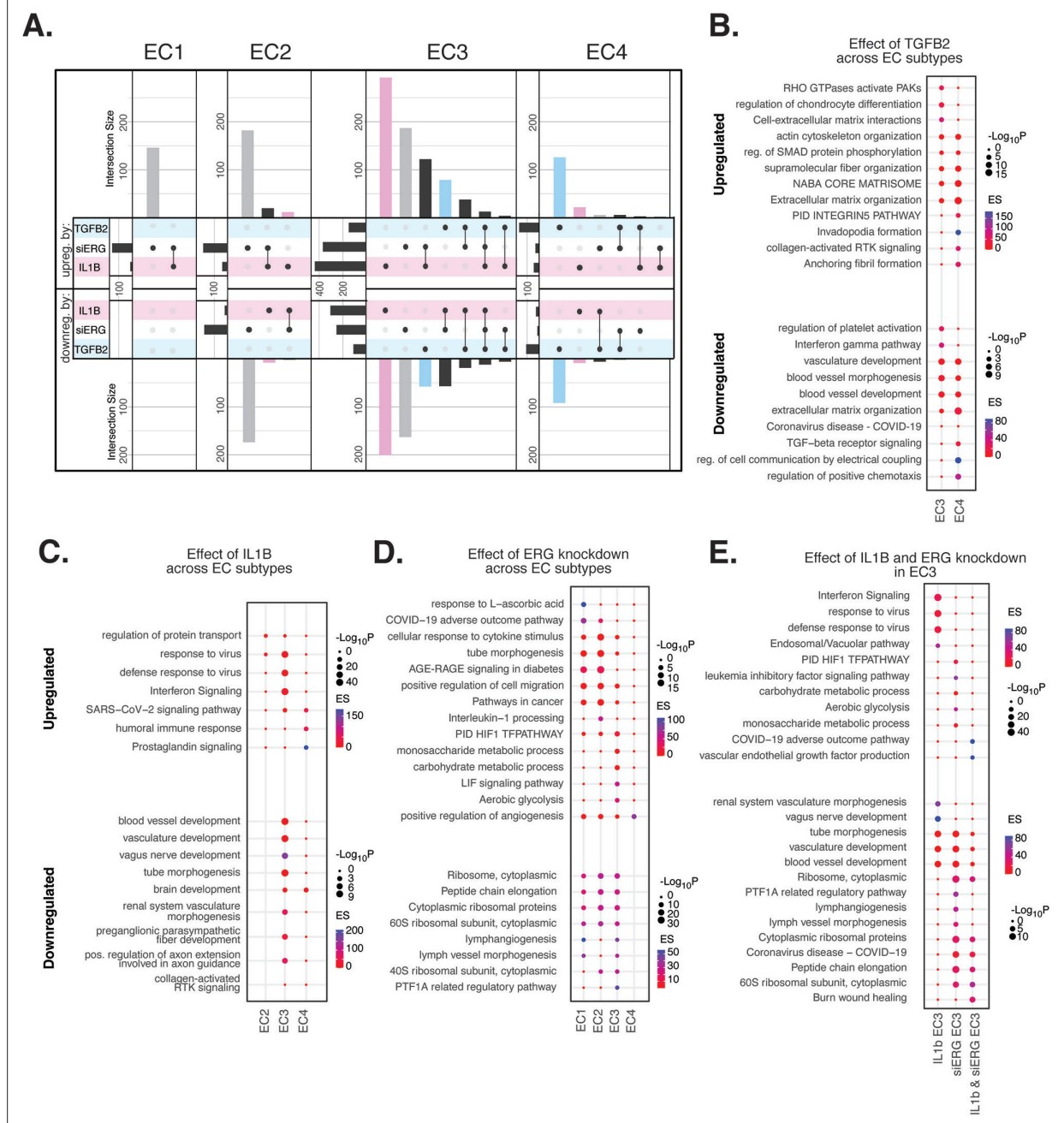

**Figure 4.** Endothelial cell (EC)-activating perturbations in vitro elicit EC subtype-specific transcriptional responses. (**A**) Upset plots of up- and downregulated differentially expressed genes (DEGs) across EC subtypes with siERG (gray), IL1B (pink), and TGFB2 (blue). Upset plots visualize intersections between sets in a matrix, where the columns of the matrix correspond to the sets, and the rows correspond to the intersections. Intersection size represents the number of genes at each intersection. (**B**) Pathway enrichment analysis (PEA) for EC3-4 up- and downregulated DEGs with TGFB2 compared to control media. (**C**) PEA for EC2-4 up- and downregulated DEGs with IL1B compared to control media. (**D**) PEA for EC1-4 up- and downregulated DEGs with siERG compared to siSCR. (**E**) PEA comparing up- and downregulated DEGs that are mutually exclusive and shared between IL1B and siERG in EC3.

The online version of this article includes the following figure supplement(s) for figure 4:

**Figure supplement 1.** Profiles of shared peaks and motif enrichments across endothelial subtypes.

Most genes significantly affected by perturbations in EC1 and EC2 were responsive to siERG, likely due to their more endothelial-like phenotypes compared to EC3 and EC4 (*Figure 4A*). siERG-affected genes in EC1 and EC2 were enriched in COVID-19 adverse outcome pathway (*Zhang et al., 2022*) (WP4891; p-values $5 \times 10^{-9}$ and $8.3 \times 10^{-5}$, for EC1-2, respectively) and AGE-RAGE signaling in diabetes (*Deng et al., 2020*) (hsa04933; p-values $8.9 \times 10^{-16}$ and $1.9 \times 10^{-20}$, respectively), while EC3 siERG-perturbed genes are enriched with a unique metabolic profile demonstrated by enrichment in monosaccharide metabolic process (GO:0005996; p-value $1 \times 10^{-6}$), carbohydrate metabolic process (GO:0005975; p-value $6.6 \times 10^{-7}$), and aerobic glycolysis (WP4629; p-value $4.1 \times 10^{-5}$) (*Figure 4D*). In contrast, EC4 siERG-induced genes are enriched in positive regulation of angiogenesis (GO:0045766; p-value $4.5 \times 10^{-6}$), a function normally impaired in *ERG*-depleted ECs (*Figure 4D*; *Fish et al., 2017*).

Due to the role that ERG plays in inhibiting NF-KB-dependent inflammation in vitro and in vivo (*Sperone et al., 2011*), we set out to characterize mutually exclusive and shared pathways between IL1B and siERG (*Figure 4E*). Importantly, siERG, but not IL1B-perturbed genes, involves several previously mentioned metabolic processes including carbohydrate metabolic process (GO:0005975; p-value $6.6 \times 10^{-7}$), aerobic glycolysis (WP4629; p-value $4.1 \times 10^{-5}$), and monosaccharide metabolic process (GO:0005996; p-value $1 \times 10^{-6}$). This suggests differences in the ability of ERG and IL1B to modify metabolism. Interestingly, IL1B but not siERG upregulated interferon signaling and viral responsive pathways (GO:0051607, p-value $1 \times 10^{-37}$; R-HSA-913531, p-value $1 \times 10^{-41}$). Shared IL1B- and siERG-upregulated genes were enriched in COVID-19 adverse outcome pathway (WP4891; p-value $1.9 \times 10^{-9}$) (*Zhang et al., 2022*). Shared IL1B- and siERG-attenuated genes are enriched in several processes involving ribosomal proteins, including ribosome, cytoplasmic (CORUM:306; p-value $3.3 \times 10^{-7}$), cytoplasmic ribosomal proteins (WP477; p-value $5.3 \times 10^{-7}$), and peptide chain elongation (R-HSA-156902; p-value $5.9 \times 10^{-7}$) (*Figure 4E*). This finding indicates that the downregulation of ribosomal genes is a hallmark of inflammatory and *ERG*-depleted endothelium. Altogether, these data demonstrate the heterogeneity in EC subtype response to pro-EndMT perturbations.

## In vitro EC EndMT models reorganize epigenetic landscapes with subtype specificity

To gain insight into gene regulatory mechanisms responsible for EC subtype transcriptional responses to IL1B, TGFB2, and siERG, we compared the effects of these perturbations on chromatin accessibility. Across all three treatments, we identified 4034 DARs (*Supplementary file 1j*, 'Materials and methods'). The majority of DARs for EC1 and EC2 were responsive to siERG, while the majority of DARs for EC3 were responsive to IL1B (*Figure 4—figure supplement 1A*, *Supplementary file 1j*). Interestingly, the epigenetic landscape of EC4 differs from its transcriptional response, insofar as most peaks were responsive to IL1B (not TGFB2) (*Figure 4—figure supplement 1A*, *Supplementary file 1j*). To inform the TFs likely bound to differentially accessible regulatory elements, motif enrichment analysis was performed (*Figure 4—figure supplement 1B–D*). Several distinct TF motifs were enriched across EC subtypes. For IL1B, we observed enrichment in KLF15 (adjusted p-value $5 \times 10^{-10}$) (Kruppel-like factor 15) in EC2 alone (*Figure 4—figure supplement 1B*). siERG-induced peaks showed subtype-specific motif enrichments, including TWIST1 (adjusted p-value $2.5 \times 10^{-22}$) (twist family bHLH transcription factor 1), HAND2 (adjusted p-value $2.3 \times 10^{-19}$) (heart and neural crest derivatives expressed 2) for EC1, RELA (adjusted p-value $9.5 \times 10^{-20}$) (RELA proto-oncogene, NF-KB subunit) for EC2, and CEBPD (adjusted p-value $1.6 \times 10^{-29}$) for EC3 (*Figure 4—figure supplement 1C*). Minimal motif enrichment was observed with siERG for EC4.

We also found several TF motifs enriched across more than one EC subtype upon perturbation. IL1B-affected peaks gained in EC1 and EC2 shared enrichments for TFDP1 (adjusted p-value $1.3 \times 10^{-4}$ and $9 \times 10^{-4}$ for EC1 and EC2, respectively) (transcription factor Dp1) and ZBTB14 motifs (adjusted p-value $2.2 \times 10^{-4}$ and $2 \times 10^{-8}$, respectively) (zinc finger and BTB domain containing 14). IL1B-induced peaks in EC3 and EC4 shared enrichment for CEBPD (adjusted p-value $4.4 \times 10^{-73}$ and $1.6 \times 10^{-33}$ for EC3 and EC4, respectively) and CEBPG motifs (adjusted p-value $5.4 \times 10^{-45}$ and $7.1 \times 10^{-18}$, respectively) (CCAAT enhancer binding protein delta and gamma) (*Figure 4—figure supplement 1B*). TGFB2-affected peaks in EC1, EC2, and EC3 shared enrichment for ZBTB14 (adjusted p-values $6.8 \times 10^{-31}$, $5.1 \times 10^{-12}$, and $2 \times 10^{-8}$, for EC1, EC2, and EC3, respectively) while TGFB2-induced peaks in EC3 and EC4 shared enrichment for the SMAD5 motif (adjusted p-value $7.4 \times 10^{-6}$ and $4.2 \times 10^{-11}$, for EC3 and EC4, respectively) (SMAD family member 5) (*Figure 4—figure supplement 1D*). Taken

together, while several enriched motifs are shared across EC subtypes, divergent epigenetic landscapes are also induced with pro-EndMT perturbations. We therefore conclude that different transcriptional responses to these perturbations across EC subtypes are elicited by distinct TFs, including members of families of the KLF, TWIST, HAND, p65, and CEBP families.

## Meta-analysis of ex vivo human atherosclerotic plaque snRNA-seq datasets

To understand the diversity of ECs in human atherosclerotic plaques and evaluate their relationships to our in vitro system, we performed a meta-analysis of data from recent publications that utilized scRNA-seq from human atherosclerotic lesions (*Pan et al., 2020*; *Alsaigh et al., 2022*; *Chowdhury et al., 2022*; *Wirka et al., 2019*; accessions in *Supplementary file 1k*). We identified 24 diverse clusters among 58,129 cells after integration of 17 different coronary and carotid samples (*Figure 5A* and *Supplementary file 1l*). Clusters were annotated using a combinatorial approach including canonical marker genes, CIPR (*Ekiz et al., 2020*), and the original publications (*Figure 5B*). Clusters were annotated as T-lymphocytes, natural killer T-cells, ECs, macrophages, VSMCs, fibroblasts, B-lymphocytes, basophils, neurons, and plasmacytoid dendritic cells (PDCs) (*Figure 5A*). We find the greatest proportion of cells belonging to each major cell type derive from carotid arteries, except for neurons that derive exclusively from coronary arteries, and PDCs that derive exclusively from carotid arteries (*Figure 5—figure supplement 1B and C*). Expected pathway enrichments are observed for annotated cell types, including NABA CORE MATRISOME (M5884; p-value $4.8 \times 10^{-41}$) for fibroblasts, blood vessel development (GO:0001568; p-value $5.6 \times 10^{-33}$) for ECs, and actin cytoskeleton organization (GO:0030036; p-value $1.3 \times 10^{-15}$) for VSMCs (*Figure 5—figure supplement 1D–G*). These observations support the diverse composition of human atherosclerotic lesions.

We evaluated what pathways distinguished the endothelial cells 1 (Endo1) and endothelial cells 2 (Endo2) subtypes from our ex vivo meta-analysis (*Figure 5C*). We found Endo2 has an EndMT-related phenotype, with enrichment in mesenchymal pathways including NABA MATRISOME ASSOCIATED (M5885; p-value $1.6 \times 10^{-14}$), ECM organization (R-HSA-1474244; p-value $6 \times 10^{-17}$), skeletal system development (GO:0001501; p-value $3.4 \times 10^{-13}$), and network map of SARS-CoV-2 signaling pathway (*Zhang et al., 2022*) (WP5115; p-value $1.3 \times 10^{-11}$) (*Figure 5C and D*). Additionally, we observe enrichment for inflammatory pathways in Endo2 including prostaglandin synthesis and regulation (WP98; p-value $1.2 \times 10^{-7}$), and complement and coagulation cascades (hsa04610; $1 \times 10^{-10}$) (*Figure 5C and D*; *Ricciotti and FitzGerald, 2011*; *Levi et al., 2004*). On the contrary, Endo1 was highly enriched in multicellular organismal homeostasis (GO:0048871; p-value $3.3 \times 10^{-8}$) and lowly enriched in mesenchymal pathways (M5885; p-value $1 \times 10^{-3}$; no enrichment for R-HSA-1474244, GO:0001501, or WP5115), indicating a canonical EC phenotype (*Figure 5C and D*). Interestingly, Endo1, but not Endo2, is highly enriched in ribosome, cytoplasmic pathway (CORUM:306; p-value $9.3 \times 10^{-96}$), and TRBP containing complex (CORUM:5380; DICER, RPL7A, EIF6, MOV10 and subunits of the 60S ribosomal particle; p-value $1.5 \times 10^{-22}$), suggesting a potential protective role for this complex along with ribosomal gene expression (*Ni and Buszczak, 2023*; *Suárez et al., 2007*). The depletion of these pathways may serve as a hallmark of activated endothelium (*Figure 5C–E*). We interpret these results to suggest that Endo1 is a classical endothelial state, while Endo2 appears to be characterized by ECM production and possibly indicate EndMT. This interpretation is further corroborated by evidence of upregulation of several classical EndMT markers in Endo2, including *FN1*, *BGN*, *COL8A1*, *ELN*, *CCN1*, and *FBLN5* (*Figure 5—figure supplement 2*; *Krizbai et al., 2015*; *Zhao et al., 2021b*; *Pinto et al., 2018*; *Stenmark et al., 2016*; *Gole et al., 2022*; *Lee et al., 2008*).

## Ex vivo-derived module score analysis reveals differences among in vitro EC subtypes and EndMT stimuli

To directly evaluate the relationships between the ex vivo and in vitro cell subpopulations, we utilized module scores. These quantitative scores are based on the sum of ex vivo marker genes across each cluster and were used to evaluate similarity to each in vitro cell subcluster. Unexpectedly, the ex vivo cluster that consistently generated the greatest module scores for in vitro ECs is the VSMCs cluster 5 (VSMC5) (*Figure 5A*, *Figure 5—figure supplement 3A*). VSMC5 bridges the EC to SMC and fibroblast clusters in the ex vivo analysis (*Figure 5A*). Marker genes of VSMC5 are expressed across ex vivo and in vitro clusters (*Figure 5—figure supplement 4A*) and include important regulators of

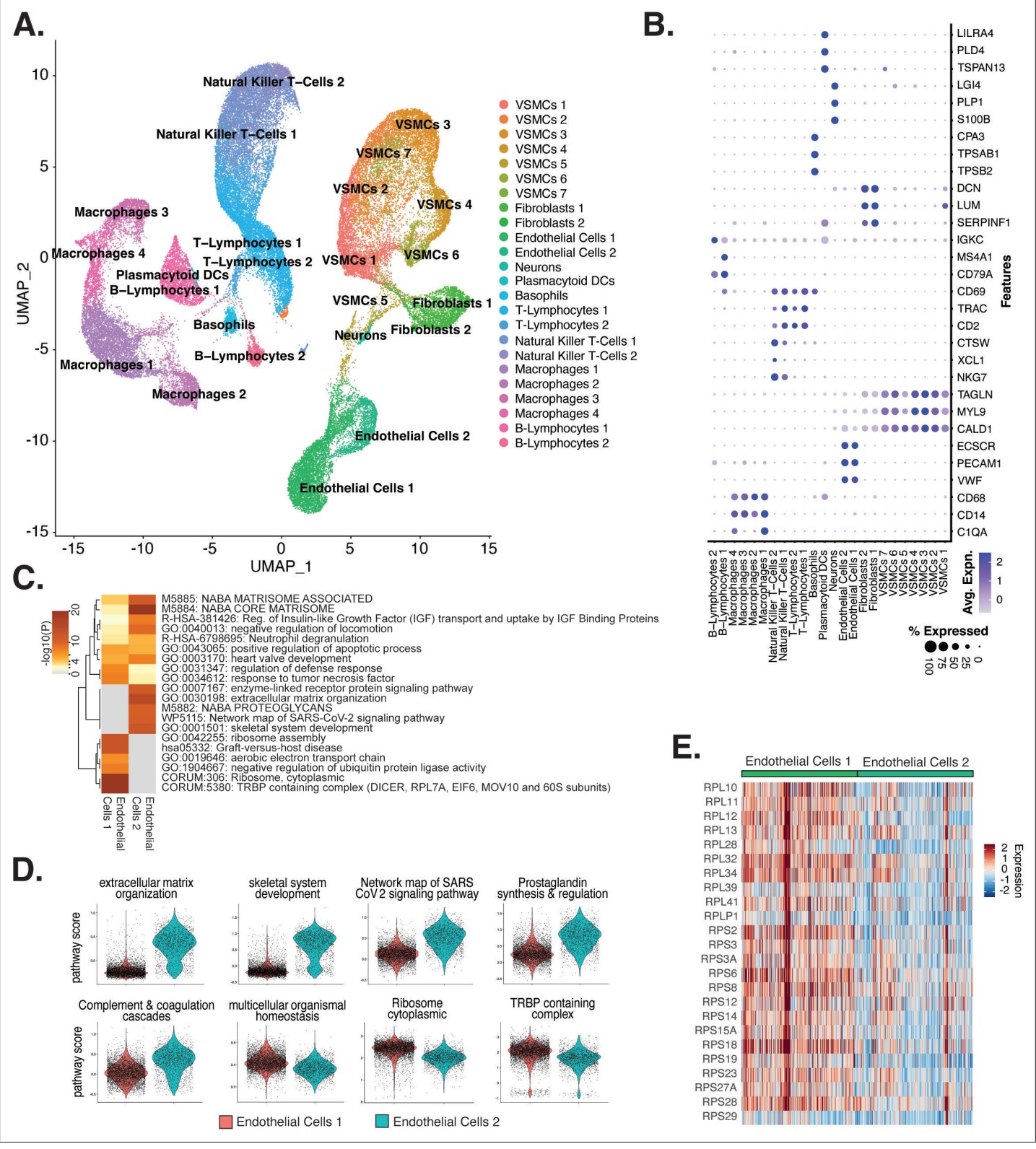

**Figure 5.** Endothelial cells (ECs) from ex vivo human atherosclerotic plaques show two major populations. (**A**) scRNA-seq UMAP of different cell subtypes across 17 samples of ex vivo human atherosclerotic plaques. (**B**) Dot plot of top markers for each cell type. (**C**) Heatmap of pathway enrichment analysis (PEA) results generated from submitting 200 differentially expressed genes (DEGs) between endothelial cells 1 (Endo1) and endothelial cells 2 (Endo2). Rows (pathways) and columns (cell subtypes) are clustered based on -Log$_{10}$(P). (**E**) Heatmap displaying expression of genes belonging to ribosome cytoplasmic pathway for Endo1 and Endo2.

*Figure 5 continued on next page*

*Figure 5 continued*

The online version of this article includes the following figure supplement(s) for figure 5:

**Figure supplement 1.** Characterization of RNA-seq profiles from ex vivo arterial samples.

**Figure supplement 2.** Violin plots displaying upregulation of several EndMT markers in Endo2, compared to Endo1, including *FN1*, *BGN*, *COL8A1*, *ELN*, *CCN1*, *FBLN5*.

**Figure supplement 3.** Cross comparisons between in vitro and ex vivo RNA-based module scores.

**Figure supplement 4.** Heatmap and pathway analysis for marker genes of the VSMC5 ex vivo cluster.

**Figure supplement 5.** Breakdown of ex vivo module scores across in vitro clusters and sample identity.

ECM, such as *BGN*, *VCAN*, *FN1*, as well as several collagen genes (*COL1A1*, *COL1A2*, *COL3A1*, *COL6A1*) (*Figure 5—figure supplement 4A and B*). VSMC5 marker transcripts also include several lncRNAs and mitochondrial transcripts (*CARMN*, *MALAT1*, *NEAT1; MT-ATP6*, *MT-ND4*, and *MT-CYB*) (*Figure 5—figure supplement 4A*). Ex vivo Endo1 and Endo2 module scores are the second highest scoring across in vitro clusters. Cells scoring high for Endo1 are concentrated in the in vitro EC1 cluster, while cells scoring high in Endo2 are concentrated to the in vitro EC3 locale (*Figure 5—figure supplement 3B–E*). This supports that EC3 is a more activated subtype than EC1. Finally, among in vitro cells, those with highest VSMC5 module scores are concentrated in EC4, underscoring that EC4 is a more mesenchymal sub-phenotype in vitro (*Figure 5—figure supplement 3B–E*).

We stratify these analyses by pro-EndMT treatment and find greater VSMC5 module scores with TGFB2 treatment versus control for EC3 (adjusted p-value=0.001) and EC4 (adjusted p-value=9.9 × 10⁻¹⁵) (*Figure 5—figure supplement 5A–C*). However, there is no difference in VSMC5 module scores for EC1-2 between control and TGFB2 treatment, suggesting these subtypes are resistant to transcriptional changes by TGFB2 exposure (i.e., EC1-2). This is in contract to the more mesenchymal-like EC (i.e., EC3-4) subtypes, which are more responsive to TGFB2 (*Figure 5—figure supplement 5A–C*, *Supplementary file 1l and m*). We observe siERG lowers Endo1 scores across all EC subtypes (adjusted p=9.9 × 10⁻¹⁵ for EC1-4), indicating *ERG* depletion decreases endothelial-likeness across all EC subtypes (*Figure 5—figure supplement 5A–C*, *Supplementary file 1m and n*). Moreover, siERG increases VSMC5 scores for EC2 (adjusted p=2.8 × 10⁻⁹) and EC3 (adjusted p-value 0.04), indicating siERG elicits activated and mesenchymal characteristics (*Figure 5—figure supplement 5A–C*, *Supplementary file 1m and n*).

## EC subtype is a major determinant in modeling cell states observed in atherosclerosis

In addition to module score analysis, we applied a complementary approach to quantitatively relate in vitro EC subtypes and pro-EndMT perturbations to ex vivo cell types. We calculate average expression profiles for all major cell populations in both ex vivo and in vitro datasets and examine the comprehensive pairwise relationship among populations with hierarchical clustering using Spearman Correlation (*Figure 6A*). All in vitro transcripts significantly regulated across all pro-EndMT perturbations at 5% false discovery rate (FDR) (*Benjamini and Hochberg, 1995*) are used in this analysis, although several additional means to select transcripts showed similar results. This analysis reveals three major observations. First, in vitro EC4 cells are most like mesenchymal ex vivo cell types, including VSMCs and fibroblasts (indicated by the yellow block of correlations in the bottom left of the heatmap in *Figure 6A*). Second, in vitro EC1, EC2, and EC3 are most like ex vivo Endo1 and Endo2 populations, especially among the siSCR and 7-day control cells. Moreover, cells in the siSCR condition in EC1 are most like ex vivo Endo1, reinforcing that these two populations are the most canonically 'healthy' endothelial populations. Third, while pro-EndMT perturbations did elicit variation in how similar in vitro ECs resembled ex vivo transcriptomic signatures, these effects are secondary to which subtype the cells belonged (*Figure 6A*). Taken together, these findings underscore that EC subtype, versus perturbation, is a greater determinant of similarity to ex vivo cell types.

## CAD-associated genetic variants are enriched across EC subtype epigenomes

Genetic predisposition to CAD is approximately 50% heritable with hundreds to thousands of genetic loci supposed to be involved in shaping an individual's propensity for disease (*Drobni et al., 2022*;

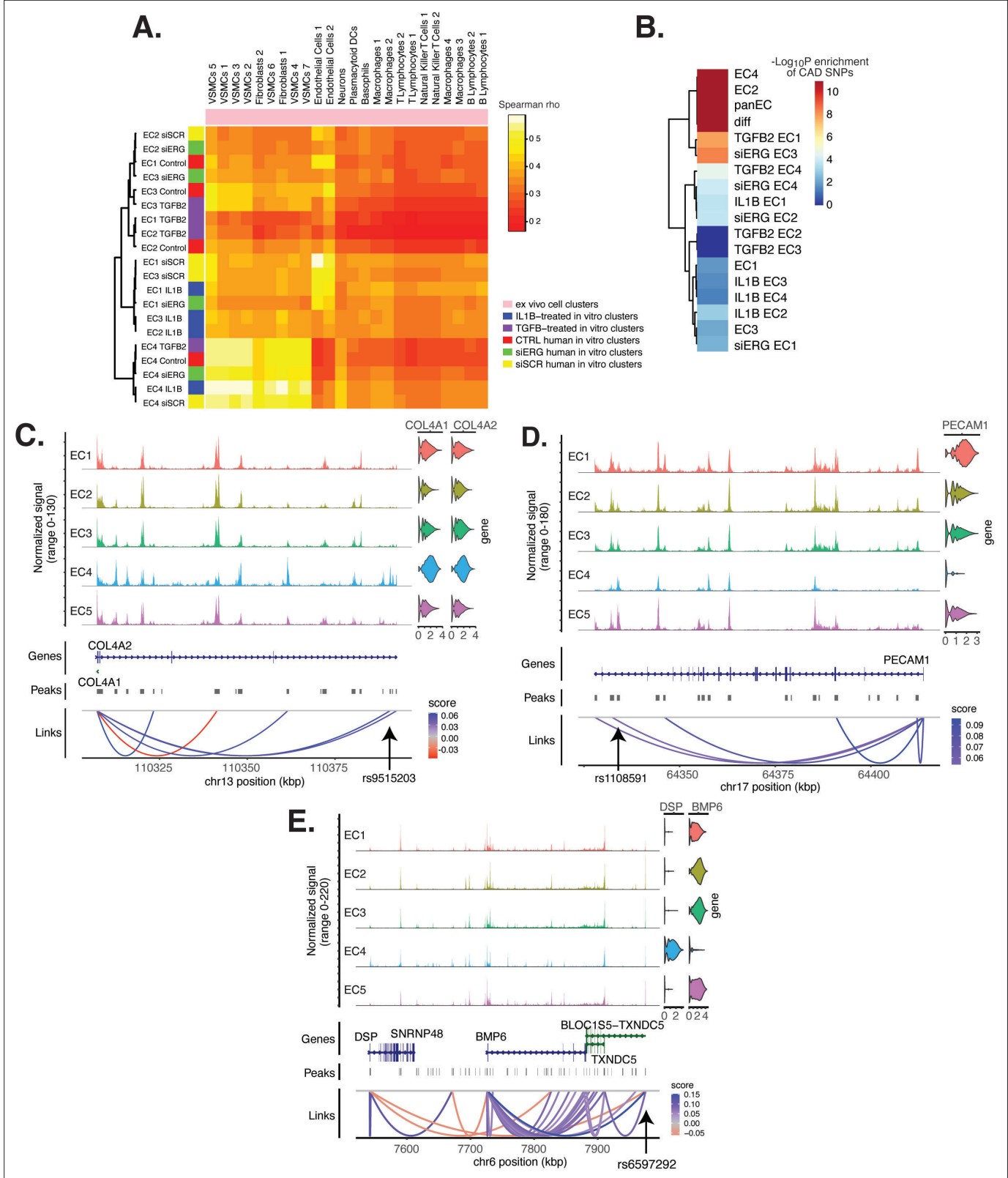

**Figure 6.** Endothelial cell (EC) subtype is a major determinant in the ability to recapitulate 'omic profiles seen in atherosclerosis. (**A**) Heatmap displaying average expression between in vitro perturbation-subtype combinations and ex vivo cell subtypes using all up- and downregulated genes between IL1B, TGFB2, or siERG versus respective controls. Spearman correlation was used as the distance metric. Rows (in vitro EC subtypes) and columns (ex vivo cell subtypes) are clustered using all significant genes (adjusted p-value<0.05) induced and attenuated across all in vitro EC subtypes for each

*Figure 6 continued*

perturbation versus its respective control. (**B**) Heatmap of coronary artery disease (CAD)-associated single-nucleotide polymorphism (SNP) enrichments across in vitro EC subtypes and perturbation–subtype combinations. Rows (EC subtypes and perturbation-subtype combinations) are clustered using -Log$_{10}$(P) for enrichment in significant CAD-associated SNPs (p-value<5 × 10$^{-8}$). Note that 'diff' represents peaks common to more than one EC subtype; it is found by subtracting EC1–5 subtype-specific peaks from the entire in vitro peak set (termed 'panEC'). (**C**) Coverage plots displaying links for *COL4A1/COL4A2* genes to EC4-specific peaks, including one overlapping with CAD-associated SNP rs9515203. (**D**) Coverage plot showing links for *PECAM1* gene to EC4-specific peaks, including one overlapping with CAD-associated SNP rs1108591. (**E**) Coverage plot showing links for *BMP6* gene to EC4-specific peaks, including one overlapping with CAD-associated SNP rs6597292.

The online version of this article includes the following figure supplement(s) for figure 6:

**Figure supplement 1.** Pathway enrichment analysis (PEA) of significant (p-value<0.05) EC4-linked genes that overlap with significant (p-value<10$^{-8}$) coronary artery disease (CAD)-associated single-nucleotide polymorphism (SNP).

*McPherson and Tybjaerg-Hansen, 2016*). Most CAD-associated variants are not protein coding, suggesting that they perturb cellular function through gene regulatory functions. We therefore asked whether the open chromatin regions in this in vitro dataset coincided with locations of single-nucleotide polymorphisms (SNPs) reported in the latest CAD meta-GWAS analysis from the Millions Veterans Project, which includes datasets from CARDIoGRAMplusC4D 1000G study, UK Biobank CAD study, and Biobank Japan (*Tcheandjieu et al., 2022*). We found significant enrichment in CAD-associated SNPs for the complete set of accessible regions across all EC subtypes (termed 'panEC'; adjusted p-value 1.5e × 10$^{-93}$; odds ratio [OR] = 1.8; *Figure 6B, Supplementary file 1o and p*) when comparing CAD SNPs exceeding the genome-wide significance threshold of p<5 × 10$^{-8}$ versus nonsignificant SNPs ('Materials and methods). Among accessible regions unique to EC subtypes, EC4 shows the greatest enrichment (adjusted p-value 7.85 × 10$^{-6}$; OR = 1.74). Additionally, EC2 is also enriched for CAD SNPs (adjusted p-value 6.3 × 10$^{-8}$; OR = 2.15), supporting a role for proliferative ECs in CAD. Of all accessible regions influenced by pro-EndMT perturbations, siERG and TGFB2 sets are most enriched for CAD variants (*Figure 6B, Supplementary file 1o and p*).

The measurement of both gene expression and DNA accessibility in the same cell enables testing for direct correlation, or 'links', between accessibility of noncoding DNA elements and gene expression of their potential regulatory targets (i.e., gene promoters). This is achieved by testing for correlation between DNA accessibility and the expression of a nearby gene across single cells (*Stuart et al., 2021*; *Cao et al., 2018*). Focusing on EC4, we search for EC4-specific sites of correlated chromatin accessibility and linked target gene expression. Upon restricting linked peaks overlapping CAD SNPs, we identify 81 significant SNP-peak-gene trios (p<0.05) representing 46 unique genes with specific activity in EC4 (*Supplementary file 1q*). We submit the 46 unique genes to Metascape (*Zhou et al., 2019*) and observe enrichment in EndMT-related pathways including blood vessel development (GO:0001568; p-value 2.1 × 10$^{-10}$), crosslinking of collagen fibrils (R-HSA-2243919; p-value 1.4 × 10$^{-8}$), and canonical and non-canonical TGFB signaling (WP3874; p-value 2.2 × 10$^{-6}$) (*Figure 6—figure supplement 1*). Literature review of this gene list further confirms several linked EC4-restricted genes associated with cardiovascular disease, including *COL4A1, COL4A2, PECAM1, DSP,* and *BMP6* (*Figure 6C–E*; *Liu et al., 2019*; *Yang et al., 2016*; *Woodfin et al., 2007*).

Altogether, these data underscore that common genetic variation influences individual propensities for CAD through ECM-organizing functions evidenced by the EC4 phenotype.

## Discussion

The major goals of this study were fourfold: (1) quantitatively assess molecular heterogeneity of cultured HAECs in vitro, (2) evaluate and compare molecular changes elicited by EC-activating perturbations at single-cell resolution, (3) assess similarities between in vitro and ex vivo EC signatures to inform the extent to which in vitro models recapitulate ex vivo biology, and (4) investigate how heterogeneous EC populations are enriched for genetic associations to CAD. The findings for each of these goals are discussed below, along with important implications and questions arising from this work.

The multiomic single-cell profiles of 15,220 cells cultured in vitro from six individuals enabled the discovery of five EC subpopulations, named EC1, EC2, EC3, EC4, and EC5. Except for EC5, EC subpopulations were comprised of cells from multiple donors and perturbations, which lends credence to the reproducibility of these biological states. The loosely defined phenotypes, based on pathway

enrichment analysis, were healthy/angiogenic for EC1, proliferative for EC2, activated for EC3, and mesenchymal for EC4. Angiogenic (*Li et al., 2019*; *Kalluri et al., 2019*; *Zhao et al., 2021a*), proliferative (*Tombor et al., 2020*; *Rodor et al., 2022*), and mesenchymal (*Tombor et al., 2020*) ECs have been previously reported in the literature. The three activating perturbations (TGFB2, IL1B, siERG) had markedly unique effects on different EC subclusters, highlighting the fact that in vitro systems contain populations of discrete cell subtypes, or states, that respond divergently to even reductionistic experimental conditions. Implications of such heterogeneity include both a need to elucidate what factors dictate treatment responsiveness, as well as experimental design and data interpretation that considers heterogeneity of response. The exact origin of EC heterogeneity observed in this study is unknown. We consider it likely that EC1 EC2, EC3, and EC4 subpopulations, which were populated by most donors, date back to the original isolation of ECs from aortic trimmings, implying that different states were preserved across passage in the culture conditions. However, we cannot exclude the possibility that some of the subpopulations have expanded since seeding of the cultures. If that were the case, EC1, EC2, EC3, and EC4 represent reproducible cell states consequent to primary culture of arterial cells. In fact, the limited correlation with ex vivo data supports this interpretation. Future studies will be required to delineate the exact source of heterogeneity in these systems.

In this study, we set out to elucidate whether the mesenchymal phenotype of EC4 was an end-stage result of EndMT and whether TGFB2, IL1B, and/or siERG would increase the proportion of cells in EC4. As shown in *Figure 3*, this hypothesis was incorrect, and the only cluster with a modest increase in cell proportions upon stimulation was EC3. Moreover, while the percent of cells in EC3 increased with TGFB or IL1B, they decreased in EC4, suggesting trans-differentiation from EC4 into EC3 with these perturbations. We cannot exclude the possibility that EC3 is an EndMT cluster, although we would have expected more significant deviation from clusters EC1 and EC2. It is also possible that the postmortem state experienced by aortic explants prior to EC isolation could induce changes in the ECs, or that the duration and doses of perturbations chosen were not sufficient to elicit complete EndMT. While the duration and doses employed in our study were established based on literature reports reporting EndMT phenotypes (*Maleszewska et al., 2013*; *Nagai et al., 2018*; *Medici et al., 2011*), EndMT was quantified by expression of only a few marker genes rather than complete transcriptomic analysis. This raises an important conclusion of our study, which is that EndMT is not well-defined molecularly and it remains possible that several different molecular profiles may each represent variant flavors of EndMT.

We found that TGFB2, IL1B, and siERG have many distinct effects on EC molecular profiles (*Figures 3 and 4*). In general, TGFB2 elicits a greater transcriptomic and epigenomic response in the mesenchymal EC subtype, EC4, while siERG and IL1B regulate the greatest numbers of shared transcripts and chromatin regions in more endothelial clusters EC1, EC2, and EC3. One interpretation for this finding is that IL1B treatment and depletion of *ERG* directly affect rewiring transcription in ECs while TGFB2 may affect other cell types in the vascular wall (or culture plate) that in turn affect ECs through paracrine interactions. Part of the similarities between IL1B and siERG responses may be explained by the fact that *ERG* depletion increases IL1B production (*Hogan et al., 2017*).

A major question raised by this work is the origin of cells in the mesenchymal cluster EC4. We originally hypothesized this cluster was the result of EndMT, which led to our investigations as to whether we could leverage EndMT-promoting exposures (IL1B, TGFB2, siERG) in vitro observe an expansion of treated cells in the EC4 population. To our surprise, the EC4 population did not expand. If anything, these exposures reduced the proportion of cells in ECs (*Figure 4*). Nonetheless, it remains a possibility that EC4 represents cells that had undergone EndMT in vivo prior to culture and that the exposures we presented in vitro were not sufficient to elicit a complete EndMT transition. Another viable hypothesis is that cells in EC4 are of SMC origin and have persisted in culture alongside their EC counterparts. Cells used in this study were isolated by luminal collagenase digestion of explanted aortic segments and were tested at early passage for EC phenotypic markers including VWF expression, cobblestone morphology, and uptake of acetylated LDL. Notably, these rigorous metrics to ensure pure EC isolation occurred prior to our group's studies. In addition, if some of the isolated cells had undergone EndMT in vivo prior to isolation, it would be nearly impossible to distinguish their cell of origin after isolation since their collective molecular phenotypes would appear as an SMC. Without lineage tracing, which is currently not possible in human tissue explants, it would not be possible to distinguish cell origin. Nonetheless, this remains an important issue that is the subject of ongoing

investigations. What we can confidently discern from these data is that these distinct cell subpopulations respond differently to the disease-relevant exposures of IL1B, TGFB2, and ERG depletion.

The current study sought to evaluate similarities and differences between in vitro primary cultures of HAECs to ex vivo single-cell signatures of cells from human lesions. First, we leveraged transcriptomic profiles from clusters in the scRNA meta-analysis of human lesions and evaluated each in vitro cluster using a module score (*Figure 5*, *Figure 5—figure supplement 4*). The three ex vivo clusters with greatest similarity to in vitro clusters were Endo1, Endo2, and VSMC5. Pathway enrichment analysis suggested that the ex vivo Endo1 cluster is close to the classic 'healthy' EC state relative to Endo2, which returned pathway enrichments consistent with activated endothelium (*Figure 5C and D*). Interestingly, Endo2 is depleted in ribosome transcripts as well as transcripts in the Dicer complex (*Figure 5C–E*), which may serve as hallmarks of dysregulated endothelium in vivo. VSMC5 is an interesting ex vivo cluster insofar as it spans the endothelial, fibroblast, and VSMC clusters (*Figure 5A*) and is enriched for genes in actin cytoskeleton, extracellular matrix organization, and more (*Figure 5—figure supplement 4*). In vitro EC1, EC2, and EC3 score generally greater in Endo1 and Endo2 relative to the more mesenchymal EC4 (*Figure 5—figure supplement 3*). Consistent with the intent of the pro-EndMT treatments, they generally decrease Endo1 and Endo2 scores and increase VSMC5 scores. However, these effects are unexceptional in comparison to effects of EC subtype. In addition to module scores, we also utilized unsupervised clustering of Spearman correlation coefficients across ex vivo and in vitro average gene expression profiles, finding again that EC1, EC2, and EC3 are more like Endo1 and Endo2 and EC4 is more like VSMCs (*Figure 6A*). As expected, the control (siSCR) cells are most correlated to healthy Endo1 transcriptomes; however, the correlation coefficient achieved is modest, at rho = 0.56. We cannot exclude the possibility that the moderate correlation coefficient observed between in vitro and ex vivo ECs may be explained by anatomic differences (i.e., aortic versus coronary and carotid arteries). While reinforcing that in vitro cell cultures best resemble ECs isolated ex vivo, regardless of perturbation, this finding accentuates how different cultured cells are and paves the way for quantitatively evaluating and improving in vitro models.

Finally, GWAS have established that hundreds of independent common genetic variants in human populations affect risk for CAD, yet discovering the causal mechanisms remains a major challenge given that most of the risk is in non-coding regions of the genome. One approach to prioritize causal variants in regulatory elements is through integration of open chromatin regions from the cell type and states of interest followed by expression quantitative trait loci (eQTL) or other linking evidence to target gene (*Stolze et al., 2020*; *Toropainen et al., 2022*). In the current study, we find significant enrichment for CAD-risk variants in open chromatin regions across the entire dataset ('panEC') as well as specifically for EC2 and EC4 subpopulations (*Figure 6B*, *Supplementary file 1o-q*). While EC3 was found to be more sensitive to perturbations in our in vitro experiments, we did not expect to see CAD-related SNPs enriched in EC3 because plasticity does not necessarily imply a pathological process. Moreover, while EC3 and EC4 both have mesenchymal phenotypes, EC3 may represent a reversible state that is lacking in EC4. This hypothesis would explain the enrichment of EC4, but not EC3, in CAD-related SNPs.

Taken together, these data emphasize the value in multimodal datasets in human samples for prioritizing disease-associated SNPs and mechanisms.

## Materials and methods
### Tissue procurement and cell culture
Primary HAECs were isolated from eight deidentified deceased heart donor aortic trimmings (belonging to three females and five males of Admixed Americans, European, and East Asian ancestries) at the University of California Los Angeles Hospital as described previously (*Navab et al., 1988*; *Supplementary file 1g*). The only clinically relevant information collected for each donor was their genotype (see 'Genotyping and multiplexing cell barcodes for donor identification'). HAECs were isolated from the luminal surface of the aortic trimmings using collagenase and identified by Navab et al. using their typical cobblestone morphology, presence of Factor VIII-related antigen, and uptake of acetylated LDL labeled with 1,1'-dioctadecyl-1–3,3,3',3'-tetramethyl-indo-carbocyan-ine perchlorate (Di-acyetl-LD) (*Navab et al., 1988*). Cells were grown in culture in M-199 (Thermo Fisher Scientific, Waltham, MA, MT-10-060-CV) supplemented with 1.2% sodium pyruvate (Thermo Fisher Scientific,

Cat# 11360070), 1% 100× Pen Strep Glutamine (Thermo Fisher Scientific, Cat# 10378016), 20% fetal bovine serum (GE Healthcare, Hyclone, Pittsburgh, PA), 1.6% Endothelial Cell Growth Serum (Corning, Corning, NY, Cat# 356006), 1.6% heparin, and 10 µl/50 ml Amphotericin B (Thermo Fisher Scientific, Cat# 15290018). Mycoplasma testing was not carried out on the cells used. HAECs at low passage (passages 3–6) were treated prior to harvest every 2 d for 7 d with either 10 ng/ml TGFB2 (Thermo Fisher Scientific, Cat# 302B2002CF), IL1B (Thermo Fisher Scientific, Cat# 201LB005CF), or no additional protein, or two doses of small interfering RNA for ERG locus (siERG; *Supplementary file 1r*), or randomized siRNA (siSCR; *Supplementary file 1r*). Donors 7 and 8 were treated prior to harvest for 6 hr with either 1 ng/ml IL1B, or no additional protein, and included in the dataset during integration to generate the original UMAP (*Figure 1B*), but not used for the purposes of downstream analyses in this study (*Supplementary file 1g*). All HAECs used were authenticated based on morphology, gene expression profiles indicative of ECs, and donor genotypes. No commercially available cell lines were analyzed in this study.

## siRNA knockdown, qPCR, and western blotting

Knockdown of ERG was performed as previously described (*Hogan et al., 2017*) using 1 nM siRNA oligonucleotides in OptiMEM (Thermo Fisher Scientific, Cat# 11058021) with Lipofectamine 2000 (Thermo Fisher Scientific, Cat# 11668030). Transfections were performed in serum-free media for 4 hr, then cells were grown in full growth media for 48 hr. All siRNAs and qPCR primers used in this study are listed in *Supplementary file 1r*. Transfection efficiency for the siRNAs utilized in this study was verified using qPCR 7 d after transfection (*Figure 3—figure supplement 3A*). Protein knockdown is shown 2 d after transfection using the same siRNAs from a representative experiment (*Figure 3—figure supplement 3B*). Antibodies used included 1:1000 recombinant anti-ERG antibody (ab133264) and 1:5000 anti-histone H3 antibody (ab1791) (Abcam). Western blots were quantified using ImageJ (*Schneider et al., 2012*).

## Nuclear dissociation and library preparation

Nuclei from primary cells were isolated according to 10X Genomics *Nuclei Isolation for Single Cell Multiome ATAC+Gene Expression Sequencing* Demonstrated Protocol (CG000365, Rev C) (*Genomics x, 2022b*). Nuclei were pooled isolated with lysis buffer consisting of 10 mM Tris-HCl (pH 7.5, Invitrogen, Cat# 15567027), 10 mM NaCl (Invitrogen, Cat# AM9759), 3 mM MgCl$_2$ (Alfa Aesar, Cat# J61014), 0.1% Tween-20 (Thermo Fisher Scientific, Cat# 9005-64-5), 0.1% IGEPAL CA-630 (Thermo Fisher Scientific, Cat# J61055.AP), 0.01% Digitonin (Thermo Fisher Scientific, Cat# BN2006), 1% BSA (Sigma-Aldrich, Cat# A2153), 1 mM DTT (Thermo Fisher Scientific, Cat# 707265 ML), 1 U/µl RNase inhibitor (Sigma Protector RNase inhibitor; Cat# 3335402001), and nuclease-free water (Invitrogen, Cat# 10977015). The seven pooled samples were incubated on ice for 6.5 min with 100 µl lysis buffer and washed three times with 1 ml wash buffer consisting of 10 mM Tris–HCl, 10 mM NaCl, 3 mM MgCl$_2$, 1% BSA, 0.1% Tween-20, 1 mM DTT, 1 U/µl RNase inhibitor, and nuclease-free water. Samples were centrifuged at 500 rcf for 5 min at 4°C, and the pellets were resuspended in chilled Diluted Nuclei Buffer consisting of 1× Nuclei Buffer (20×) (10X Genomics), 1 mM DTT (Thermo Fisher Scientific, Cat# 707265 ML), 1 U/µl RNase inhibitor, and nuclease-free water. The homogenate was filtered through a 40 µm cell strainer (Flowmi, Cat# BAH136800040) prior to proceeding immediately to 10X Chromium library preparation according to the manufacturer's protocol (CG000338).

## Genotyping and multiplexing cell barcodes for donor identification

Genotyping of HAEC donors was performed as described previously (*Stolze et al., 2020*). Briefly, IMPUTE2 (*Howie et al., 2009*) was used to impute genotypes utilizing all populations from the 1000 Genomes Project reference panel (phase 3) (*1000 Genomes Project Consortium et al., 2015*). Genotypes were called for imputed SNPs with allelic R2 values greater than 0.9. Mapping between genomic coordinates was performed using liftOver (*Kuhn et al., 2013*). VCF files were subset by genotypes for the donors of interest using VCFtools (*Danecek et al., 2011*).

To identify donors across the in vitro dataset, snATAC- and snRNA-seq output BAM files from Cell Ranger ARC (10X Genomics, v.2.0.0; *Genomics x, 2022a*) were concatenated, sorted, and indexed using samtools (*Danecek et al., 2021*). The concatenated BAM files were input with the genotype VCF file to demuxlet (*Kang et al., 2018*) to identify best matched donors for each cell barcode, using

options '–field GT'. Verification of accurate donor identification was confirmed by visualizing female sex-specific *XIST* for the known donor sexes (*Figure 1—figure supplement 2*).

## snRNA-seq bioinformatics workflow

A target of 10,000 nuclei were loaded onto each lane. Libraries were sequenced on NovaSeq6000. Reads were aligned to the GRCh38 (hg38) reference genome and quantified using Cell Ranger ARC (10X Genomics, v.2.0.0; *Genomics x, 2022a*). Datasets were subsequently preprocessed for RNA individually with Seurat version 4.3.0 (*Hao et al., 2021*). Seurat objects were created from each dataset, and cells with <500 counts were removed. This is a quality control step as it is thought that cells with low number of counts are poor data quality. Similarly, for each cell, the percentage of counts that come from mitochondrial genes was determined. Cells with >20% mitochondrial gene percent expression (which are thought to be of low quality, possibly due to membrane rupture) were excluded. Demuxlet (*Kang et al., 2018*) was next used to remove doublets. The filtered library was subset and merged by pro-EndMT perturbation. Data were normalized with NormalizeData, and cell cycle regression was performed by generating cell cycle phase scores for each cell using CellCycleScoring, followed by regression of these using ScaleData (*Luecken and Theis, 2019*). Batch effects by treatment were corrected using FindIntegrationAnchors using 10,000 anchors, followed by IntegrateData.

## snATAC-seq bioinformatics workflow

A target of 10,000 nuclei were loaded onto each lane. Libraries were sequenced on an NovaSeq 6000 according to manufacturer's specifications at the University of Chicago. Reads were aligned to the GRCh38 (hg38) reference genome and quantified using Cell Ranger ARC (10X Genomics, v.2.0.0; *Genomics x, 2022a*). Datasets were subsequently preprocessed for ATAC individually with Seurat v4.3.0 (*Hao et al., 2021*) and Signac v1.6.0 (*Heidecker et al., 2010*) to remove low-quality nuclei (nucleosome signal >2, transcription start site enrichment <1, ATAC count <500, and % mitochondrial genes >20) (*Hao et al., 2021*). Next, demuxlet (*Kang et al., 2018*) was used to remove doublets. A common peak set was quantified across snATAC-seq libraries using FeatureMatrix, prior to merging each lane. A series of two iterative peak calling steps were performed. The first step consisted of calling peaks for every EndMT perturbation, and the second involved calling peaks for every cluster generated from weighted nearest-neighbor analysis (WNN) (see 'Integration and weighted nearest-neighbor analyses'). Latent semantic indexing (LSI) was computed after each iterative peak calling step using Signac standard workflow (*Stuart et al., 2021*). Batch effects by treatment were finally corrected using FindIntegrationAnchors using 10,000 anchors, followed by IntegrateData.

## Integration and weighted nearest-neighbor analyses

Following snRNA-seq and snATAC-seq quality control filtering, barcodes for each modality were matched, and both datasets were combined by adding the snATAC-seq assay and integrated LSI to the snRNA-seq assay. WNN (*Hao et al., 2021*) was next calculated on the combined dataset, followed by joint UMAP (_WNN_UMAP) visualization using Signac (*Stuart et al., 2021*) functions FindMultimodalNeighbors, RunUMAP, and FindClusters, respectively. WNN is an unsupervised framework to learn the relative utility of each data type in each cell, enabling an integrative analysis of multimodal datasets. This process involves learning cell-specific modality 'weights' and constructing a _WNN_UMAP that integrates the modalities. The subtypes discovered in the first round of WNN were utilized in an additional peak calling step for snATAC-seq, followed by LSI computation, re-integration, and a final round of WNN to achieve optimal peak predictions (see 'Single-Nucleus ATAC sequencing bioinformatics workflow') (*Yan et al., 2020*).

## Differential expression and accessibility region analyses across EC subtypes and EndMT perturbation–subtype combinations

Differential expression between clusters was computed by constructing a logistic regression (LR) model predicting group membership based on the expression of a given gene in the set of cells being compared. The LR model included pro-EndMT perturbation as a latent variable and was compared to a null model using a likelihood ratio test. This was performed using Seurat FindMarkers, with 'test.use=LR' and 'latent.vars' set to perturbation. Differential expression between perturbation and control for each cluster was performed using pseudobulk method with DESeq2 (*Love et al., 2014*).

Raw RNA counts were extracted for each EndMT perturbation-subtype combination and counts, and metadata were aggregated to the sample level.

Differential accessibility between EC subtypes was performed using FindMarkers, with 'test.use= LR' and latent.vars set to both the number of reads in peaks and perturbation. Finally, differential accessibility between perturbation and control for each cluster was performed using FindMarkers, with 'test.use=LR' and latent.vars set to the number of reads in peaks.

Bonferroni-adjusted p-values were used to determine significance at adjusted p-value<0.05 for differential expression, and p-value<0.005 for differential accessibility (*Benjamini and Hochberg, 1995*).

### Pathway enrichment analysis

Pathway enrichment analysis (PEA) was performed using Metascape (*Zhou et al., 2019*). Top DEGs for each EC subtype or subtype–perturbation were sorted based on ascending p-value. Genes listed for each pathway were pulled from the Metacape results file, '_FINAL_GO.csv'. For heatmaps produced by metascape, top 20 or 100 pathways were pulled from Metascape.png files, 'HeatmapSelectedGO. png', 'HeatmapSelectedGOParent.png', or 'HeatmapSelectedGOTop100.png'.

### Motif enrichment analysis

A hypergeometric test was used to test for overrepresentation of each DNA motif in the set of differentially accessible peaks compared to a background set of peaks. We tested motifs present in the Jaspar database (2020 release) (*Fornes et al., 2020*) by first identifying which peaks contained each motif using motifmatchr R package (https://bioconductor.org/packages/motifmatchr). We computed the GC content (percentage of G and C nucleotides) for each differentially accessible peak and sampled a background set of 40,000 peaks matched for GC content (*Stuart et al., 2021*). Per-cell motif activity scores were computed by running chromVAR (*Schep et al., 2017*), and visualized using Seurat (*Hao et al., 2021*) function FeaturePlot.

### Human atherosclerosis scRNA-seq public data download, mapping, and integration across samples

Count matrices of 17 samples taken from four different published scRNA-seq datasets were downloaded from the NCBI Gene Expression Omnibus (accessions listed in *Supplementary file 1k*), processed using Cell Ranger (10X Genomics Cell Ranger 6.0.0; *Zheng et al., 2017*) with reference GRCh38 (version refdata-gex-GRCh38-2020-A, 10X Genomics), and analyzed using Seurat version 4.3.0 (*Hao et al., 2021*). Seurat objects were created from each dataset, and cells with <500 counts and >20% mitochondrial gene percent expression were excluded. Additionally, doublets were removed using DoubletFinder (*McGinnis et al., 2019*), which predicts doublets according to each real cell's proximity in gene expression space to artificial doublets created by averaging the transcriptional profile of randomly chosen cell pairs. Next, normalization and variance stabilization, followed by PC analysis for 30 PCs, were performed in Seurat (*Hao et al., 2021*) using default parameters. Batch effects across the 17 samples were corrected using Seurat functions (*Hao et al., 2021*) FindIntegrationAnchors using 10,000 anchors, followed by IntegrateData. During the integration step, cell cycle regression was performed by assigning cell cycle scores with Seurat (*Hao et al., 2021*) function CellCycleScoring. The ex vivo dataset was first visualized, and canonical markers were identified for annotating cell types using FindAllMarkers.

### Module scoring

FindAllMarkers was used to identify the top DEGs between each ex vivo cell subtype. Cells from the in vitro dataset were assigned an ex vivo cell subtype module score using Seurat (*Hao et al., 2021*) function AddModuleScore. The difference in module score between each in vitro EC subtype was established using Wilcoxon rank sum test with continuity correction and a two-sided alternative hypothesis.

### Comparison of ex vivo snRNA-seq data to in vitro snRNA-seq data

Meta-analyzed ex vivo human scRNA-seq data and in vitro snRNA-seq data were compared. Gene expression values for each ex vivo cell subtype and in vitro EC subtype–perturbation were produced using the AverageExpression function in Seurat (*Hao et al., 2021*) (which exponentiates log data,

therefore output is depth normalized in non-log space). *Figure 6A* was generated using hclust function in R (*Murtagh and Legendre, 2014*). Spearman correlation was used as the distance metric. Sample clustering was performed using all significant genes (adjusted p-value <0.05) induced and attenuated across all in vitro EC subtypes for each pro-EndMT perturbation versus its respective control. *Figure 5—figure supplement 4A* was made using average expression data for marker genes for each ex vivo cell subtype. Hierarchical clustering across ex vivo cell subtypes was performed using hclust function in R (*Murtagh and Legendre, 2014*) using average expression as the distance metric for a given gene.

## GWAS SNP enrichment analysis

The SNPs associated with CAD were extracted from the most recent available meta-analysis (*Tcheandjieu et al., 2022*). We utilized a matched background of SNPs pulled from 1000 Genomes Project reference panel (phase 3) (*1000 Genomes Project Consortium et al., 2015*), which were filtered using PLINK (*Purcell et al., 2007*) v1.90b5.3 with the following settings: '--maf 0.01', '--geno 0.05'. Mapping between genomic coordinates was performed using liftOver (*Kuhn et al., 2013*). To evaluate for enrichment in CAD-associated SNPs for each EC subtype and perturbation-subtype peak set, traseR package in R (traseR) (*Chen and Qin, 2016*) was used with the following: 'test.method' = 'fisher', 'alternative' = 'greater'.

## Peak-to-gene linkage

We estimated a linkage score for each peak-gene pair using the LinksPeaks function in Signac (*Stuart et al., 2021*). For each gene, we computed the Pearson correlation coefficient $r$ between the gene expression and the accessibility of each peak within 500 kb of the gene TSS. For each peak, we then computed a background set of expected correlation coefficients given properties of the peak by randomly sampling 200 peaks located on a different chromosome to the gene, matched for GC content, accessibility, and sequence length (MatchRegionStats function in Signac). We then computed the Pearson correlation between the expression of the gene and the set of background peaks. A z score was computed for each peak as $z = (r - \mu)/\sigma$, where $\mu$ is the background mean correlation coefficient and $\sigma$ is the SD of the background correlation coefficients for the peak. We computed a p-value for each peak using a one-sided z-test and retained peak-gene links with a p-value<0.05 and a Pearson correlation coefficient. The results were restricted to peak regions that overlapped with significant CAD-associated SNPs (see 'GWAS SNP enrichment analysis').

## Data visualization

Data visualizations were performed using Seurat functions DimPlot, DotPlot, FeaturePlot, and VlnPlot. Other data visualizations were performed using ggplot2 (for stacked bar graphs) (*Villanueva and Chen, 2019*), UpSetR (for UpSet plots) (*Conway et al., 2001*), pheatmap (for DEG and DAR analysis heatmaps), and heatmap.2 (for Spearman's rank correlation coefficient heatmap and *Figure 5—figure supplement 4A*; *Warnes et al., 2016*).

## Acknowledgements

This study was funded by grants from the National Institutes of Health through R01HL147187 (CER), R35GM137896 (DAC), F30HL162469 (MLA), T32HL7249-45 (MLA), and the Geneen Charitable Trust Awards Program for Coronary Heart Disease Research (CER).

## Additional information

### Funding

| Funder | Grant reference number | Author |
| --- | --- | --- |
| National Institutes of Health | R01HL147187 | Casey E Romanoski |

| Funder | Grant reference number | Author |
| --- | --- | --- |
| National Institutes of Health | R35GM137896 | Darren A Cusanovich |
| National Institutes of Health | F30HL162469 | Maria L Adelus |
| National Institutes of Health | T32HL7249-45 | Maria L Adelus |

The funders had no role in study design, data collection and interpretation, or the decision to submit the work for publication.

## Author contributions

Maria L Adelus, Conceptualization, Formal analysis, Funding acquisition, Validation, Visualization, Writing – original draft, Writing – review and editing; Jiacheng Ding, Methodology; Binh T Tran, Formal analysis; Austin C Conklin, Supervision, Methodology; Anna K Golebiewski, Resources, Writing – review and editing; Lindsey K Stolze, Michael B Whalen, Resources; Darren A Cusanovich, Supervision, Methodology, Writing – review and editing; Casey E Romanoski, Conceptualization, Resources, Formal analysis, Supervision, Visualization, Methodology, Writing – original draft, Writing – review and editing

## Author ORCIDs

Maria L Adelus ⓘ https://orcid.org/0000-0002-9676-9214
Darren A Cusanovich ⓘ https://orcid.org/0000-0001-6889-0095
Casey E Romanoski ⓘ http://orcid.org/0000-0002-0149-225X

Reviewer #2 (Public review): https://doi.org/10.7554/eLife.91729.3.sa1
Author response https://doi.org/10.7554/eLife.91729.3.sa2

---

# Additional files

## Supplementary files

• Supplementary file 1. (a) The numbers of total reads and reads per nucleus in snRNA-seq and snATAC-seq data: total reads and reads per nucleus in snRNA-seq and snATAC-seq data are shown. (b) Cells and transcripts detected per nucleus for each snRNA-seq library after filtering during quality control. (c) Proportion of cells belonging to each donor for each EC cluster. (d) Markers for each cluster: discovered using Seurat's FindAllMarkers function with latent.vars set to 'Treatment'. Each gene is tested for differential expression between cells of each cluster and the cells outside of that cluster. p_val corresponds to the p-value of the Wilcoxon rank sum test, avg_log2FC is the log2-fold change in expression of the marker for the average cell in the cluster. Subtype is the designated subtype chosen for each cluster, cluster is the cluster of cells for which the marker was discovered, gene is the markers, pct.1 is the percentage of cells expressing the marker within the tested cluster, pct.2 is the percentage of cells expressing the marker outside the cluster, and p_val_adj is the Bonferroni adjusted p-value (based on the number of genes tested). (e) Top biological pathways from Metascape pathway enrichment analysis (and corresponding p-values) taken from top 100 significant genes for each cluster from FindAllMarkers. (f) Cells and peaks detected per nucleus for each snATAC-seq library after filtering during quality control. (g) List of chemical and genetic perturbations that each primary cell donor underwent: TGFB2=7-day TGFB2 (10 ng/ml); IL1B=7-day IL1B (10 ng/ml); IL1B6h=6 hr IL1B (1 ng/ml); siERG = siRNA-mediated knock-down of ERG (siERG), achieved with serial transfections on days 0 and 4. Ancestries were determined according to 1000 Genomes Project superpopulations. (h) Differential accessible regions (p<0.005) across EC subtypes. (i) Differentially expressed genes (DEGs) with TGFB2, IL1B, or siERG induction across EC subtypes: briefly, pseudobulk DEG analysis is performed to find genes that are significantly differentially expressed between perturbations and respective controls for pan ECs (EC1-4) and each EC subtype. DEGs with adjusted p-value<0.05 are considered significant and listed below. (j) Differentially accessible regions affected by TGFB2, IL1B, or siERG across EC subtypes: briefly, significant DARs were found using FindMarkers using logistic regression with 'nCount_ATAC' set as a latent variable. DARs with p<0.005 were considered significant and listed below. (k) Description of public data used in this study. (l) Markers for each cluster: discovered using Seurat's FindAllMarkers function; briefly,

each gene is tested for differential expression between cells of each cluster and the cells outside of that cluster. p_val corresponds to the p-value of the Wilcoxon rank sum test, avg_log2FC is the log2-fold change in expression of the marker for the average cell in the cell type, cell type is the collapsed cluster of cells for which the marker was discovered, gene is the markers, pct.1 is the percentage of cells expressing the marker within the tested cluster, pct.2 is the percentage of cells expressing the marker outside the cluster, and p_val_adj is the Bonferroni adjusted p-value (based on the number of genes tested). (m) Effect of EndMT perturbation on in vitro EC subtypes according to ex vivo cell type module scores: Adjusted p-values<0.05 are generated using Wilcoxon rank sum test with continuity correction setting alternative hypothesis to 'two.sided'. Colored arrows represent significantly (adjusted p-value<0.05) upregulated (green) and downregulated (red) module scores for each EC sub-phenotype and perturbation combination. (n) Results table for effect of EndMT perturbation on in vitro EC subtypes according to ex vivo cell type module scores: Adjusted p-values<0.05 are generated using Wilcoxon rank sum test with continuity correction setting alternative hypothesis to 'two.sided'. (o) SNP enrichment analysis results across clusters and perturbation-cluster combinations: briefly, SNP enrichment analysis was used with traseR with 'test.method' set to 'fisher' and 'alternative' set to 'greater'. (p) Significantly enriched SNPs (adjusted p-value<0.05) across clusters and perturbation-cluster combinations: briefly, SNP enrichment analysis was used with traseR with 'test.method' set to 'fisher' and 'alternative' set to 'greater'. (q) Significant links between genes and EC4-specific peaks which overlap with CAD-associated SNPs (p-value<0.05): Briefly, peak-to-gene linkage was performed using Signac 'LinkPeaks' function on a dataset consisting of EC4-specific peaks and genes. Results were filtered based on peak regions which overlap with significant (p<5e-8) CAD associated SNPs. (r) siRNAs and qPCR primers used in this study. siERG #1, #2, #4, and #5 were pooled together. Non-targeting siRNA (siSCR) #3 and #4 were pooled together.

- MDAR checklist

## Data availability

Sequencing data have been deposited in GEO under accession code GSE228428. This project utilized data deposited previously in GEO accessions GSE155512, GSE159677, and GSE131778. The code used for analysis can be found in GitHub at https://github.com/cromanoski/Adelus_2024_Elife/ (copy archived at *Romanoski, 2024*).

The following dataset was generated:

| Author(s) | Year | Dataset title | Dataset URL | Database and Identifier |
|---|---|---|---|---|
| Adelus ML, Ding J, Tran BT, Conklin AC, Golebiewski AK, Stolze LK, Whalen MB, Cusanovich DA, Romanoski CE | 2023 | Multiomic profiling of in vitro models of endothelial-to-mesenchymal transition reveals endothelial cell subtype is a major determinant of fidelity to observed states in atherosclerosis | https://www.ncbi.nlm.nih.gov/geo/query/acc.cgi?acc=GSE228428 | NCBI Gene Expression Omnibus, GSE228428 |

The following previously published datasets were used:

| Author(s) | Year | Dataset title | Dataset URL | Database and Identifier |
|---|---|---|---|---|
| Huize P, Chenyi X | 2020 | Single-cell genomics reveals a novel cell state during smooth muscle cell phenotypic switching and potential therapeutic targets for atherosclerosis in mouse and human | https://www.ncbi.nlm.nih.gov/geo/query/acc.cgi?acc=GSE155512 | NCBI Gene Expression Omnibus, GSE155512 |
| Alsaigh T, Evans D, Frankel D, Torkamani A | 2020 | Decoding the transcriptome of calcified atherosclerotic plaque at single-cell resolution | https://www.ncbi.nlm.nih.gov/geo/query/acc.cgi?acc=GSE159677 | NCBI Gene Expression Omnibus, GSE159677 |

*Continued*

| Author(s) | Year | Dataset title | Dataset URL | Database and Identifier |
|---|---|---|---|---|
| Wirka RC, Wagh D, Paik DT, Pjanic M, Nguyen T, Miller CL, Kundu R, Nagao M, Coller J, Koyano T, Fong R, Woo YJ, Liu B, Montgomery SB, Zhu K, Chang R, Alamprese M, Tallquist MD, Kim JB, Quertermous T, Wu J | 2019 | Single cell analysis of smooth muscle cell phenotypic modulation in vivo during disease in mice and humans | https://www.ncbi.nlm.nih.gov/geo/query/acc.cgi?acc=GSE131778 | NCBI Gene Expression Omnibus, GSE131778 |

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
