## [Editor Report · eLife assessment]

This is a **fundamental** resource of snRNA-seq and chromatin accessibility data from human aortic endothelial cells (ECs), treated with relevant perturbations such as IL1b, TGFB2, or siERG. The authors show that ECs can be categorized by distinct subpopulations of differing plasticity. The support for the existence of these subpopulations is **compelling**, supported also by three publicly available scRNA-seq datasets, and differential enrichment of coronary artery disease associated SNPs in open chromatin in these subpopulations.

---

## [Referee Report · Reviewer #2 (Public review)]

This study by Adelus et al. profiled the transcriptome and chromatin accessibility in cultured human aortic endothelial cells (ECs) at single-cell resolution. They also stimulated these cells with EC-activating agents, such as IL1b, TGFB2, or siEGR, to knock down this master transcription factor in ECs. The results show a subpopulation, EC3, with the highest plasticity and sensitivity to perturbations. The authors also reviewed and meta-analyzed three independent publicly available scRNA-seq datasets, identifying two distinct EC subpopulations. Additionally, they aligned CAD-related SNPs with open chromatin regions in EC subpopulations. This study provides fundamental evidence to enrich our understanding of vascular ECs and highlights potential subpopulations that may contribute to health and diseases. The work exhibits the potential impact in the field.

Comments on revised version:

I appreciate their revision, which addressed all my concerns. I understand the current technique's limitation in distinguishing bona fide cell lineages from human tissue explants, but it merits further investigation. This is because EC4 may also be involved in critical pathological processes. Again, this work established a solid foundation for exploring endothelial cell plasticity.

---

## [Author Response]

The following is the authors’ response to the original reviews.

**Reviewer #1 (Public Review):**
Summary:The manuscript by Adelus and colleagues investigates the snRNA sequencing of endothelial cells isolated from deceased heart donor aortic trimmings. From n=6 donors, the authors have identified 5 distinct endothelial cell (EC) populations. The expression levels of a set of genes are different among the different donors and different EC clusters. Furthermore, treatment with IL1B, TGFB, or ERGsi decreased the proportion of some of these clusters and increased others, with some migratory and ECM-producing capacity. Another interesting observation in this study is that IL-1B alone induces a shift in the clusters and that is different from the TGFB-induced cells. However, ex vivo analyses showed most of the TGFB-induced population matched the in vitro observations. Another interesting finding of the work is that the authors detected SNPs linked to chromatin accessibility to the set of genes identified within these EC populations.Strengths:Overall, the work is intriguing and has some novel aspects to it, especially the link between ECderived EndMT in culture and comparing that with ex vivo atherosclerotic samples.

In summary, we thank we thank Reviewer #1 for raising questions that prompted new speculations and clarifications of our data. We hope this Reviewer will now find our revised manuscript suitable for publication.

Weaknesses:The experiments are lacking in controls, the purity of the isolation, and the use of multiple donors (deceased hearts) to draw conclusions. The lack of validation of the work is a concern.

We thank Reviewer #1 for raising these concerns. Controls were not available in the public in vivo data, likely due to the systemic nature of coronary artery disease (CAD) and the logistical difficulty in obtaining arterial samples from healthy participants. With respect to our in vitro data, controls were included in the design. We agree that it is critical to validate functions of endothelial cell (EC) populations with functional studies, and this is the subject of ongoing and future work. Regarding asymmetry of donors, we aimed to have at least three replicate donors per condition. In the study design, we had to load genetically different donors per 10x lane, which is why we utilized different donors for each condition. We address the purity of isolation in our response to Reviewer #2 below.

**Reviewer #2 (Public Review):**
This study by Adelus et al. profiled the transcriptome and chromatin accessibility in cultured human aortic endothelial cells (ECs) at single-cell resolution. They also stimulated these cells with EC-activating agents, such as IL1b, TGFB2, or siEGR, to knock down this master transcription factor in ECs. The results show a subpopulation, EC3, with the highest plasticity and sensitivity to perturbations. The authors also reviewed and meta-analyzed three independent publicly available scRNA-seq datasets, identifying two distinct EC subpopulations. Additionally, they aligned CAD-related SNPs with open chromatin regions in EC subpopulations. This study provides fundamental evidence to enrich our understanding of vascular ECs and highlights potential subpopulations that may contribute to health and diseases. The work exhibits the potential impact in the field. While the manuscript is comprehensive, there are some concerns that should be addressed.(1) My major concern is whether EC4 is derived from ECs. It seems that EC4 showed a lesser reaction to those perturbations and had lower expression levels of EC marker genes. Did the authors evaluate the purity of their isolated HAECs? Please discuss the potential cell lineage mapping of EC4.

We thank Reviewer #2 for raising the question on the purity of isolation. We have now included this in the Discussion:

“A major question raised by this work is the origin of cells in the mesenchymal cluster EC4. We originally hypothesized this cluster was the result of EndMT, which led to our investigations as to whether we could leverage EndMT-promoting exposures (IL1B, TGFB2, siERG) in vitro observe an expansion of treated cells in the EC4 population. To our surprise, the EC4 population did not expand. If anything, these exposures reduced the proportion of cells in ECs (Figure 4). Nonetheless, it remains a possibility that EC4 represents cells that had undergone EndMT in vivo prior to culture and that the exposures we presented in vitro were not sufficient to elicit a complete EndMT transition. Another viable hypothesis is that cells in EC4 are of SMC origin and have persisted in culture alongside their EC counterparts. Cells used in this study were isolated by luminal collagenase digestion of explanted aortic segments and were tested at early passage for EC phenotypic markers including VWF expression, cobblestone morphology, and uptake of acetylated LDL. Notably, these rigorous metrics to ensure pure EC isolation occurred prior to our group’s studies. In addition, if some of the isolated cells had undergone EndMT in vivo prior to isolation, it would be nearly impossible to distinguish their cell of origin after isolation since their collective molecular phenotypes would appear as an SMC. Without lineage tracing, which is currently not possible in human tissue explants, it would not be possible to distinguish cell origin. Nonetheless, this remains an important issue that is the subject of ongoing investigations. What we can confidently discern from these data is that these distinct cell subpopulations respond differently to the disease-relevant exposures of IL1B, TGFB2, and ERG depletion.”

(2) Although all the donors are de-identified, is there any information about the severity of their vascular impairment, particularly in the case of patient 5, who exhibits the unique EC5?

All donors are de-identified, and we only have access to their genotypes. We have now clarified this in Methods, “Tissue Procurement and Cell Culture”:” Primary HAECs were isolated from eight de-identified deceased heart donor aortic trimmings (belonging to three females and five males of Admixed Americans, European, and East Asian ancestries) at the University ofCalifornia Los Angeles Hospital as described previously (42) (Table S7 in the Data Supplement).The only clinically relevant information collected for each donor was their genotype (Methods, “Genotyping and Multiplexing Cell Barcodes for Donor Identification”).”

(3) The meta-analysis of the published datasets is comprehensive. The identified EC heterogeneity corresponds to their in vitro data. I am wondering, in terms of transcriptome, is there any similarity between endo1 and EC1/EC2, and also endo2 and EC3/EC4?

This was addressed in Results, “Ex Vivo-derived Module Score Analysis Reveals Differences among In Vitro EC Subtypes and EndMT Stimuli”: “Cells scoring high for Endo1 are concentrated in the in vitro EC1 cluster, while cells scoring high in Endo2 are concentrated to the in vitro EC3 locale (Figure S7B-E in the Data Supplement).”

(4) The in vitro data indicates that EC3 shows the highest plasticity and sensitivity to perturbations, which may act as the major subtype of ECs responding to risk factors. It's very interesting that CAD-related SNPs do not seem to be enriched in EC3. Please discuss this discrepancy.

We thank Reviewer #2 for bringing up this interesting point, which we have now included in our Discussion: “While EC3 was found to be more sensitive to perturbations in our in vitro experiments, we did not expect to see CAD-related SNPs enriched in EC3 because plasticity does not necessarily imply a pathological process. Moreover, while EC3 and EC4 both have mesenchymal phenotypes, EC3 may represent a reversible state that is lacking in EC4. This hypothesis would explain the enrichment of EC4, but not EC3, in CAD-related SNPs.”

(5) The last sentence in the legend of Figure 1 seems incomplete: 'Module scores are generated for each cell barcode with Seurat function AddModuleScore().'

We have made changes to this sentence so that it now reads: “Module scores are generated for each cell barcode with the Seurat function AddModuleScore().”

**Recommendations for the authors:**

**Reviewer #1 (Recommendations For The Authors):**
The manuscript by Adelus and colleagues investigates the snRNA sequencing of endothelial cells isolated from deceased heart donor aortic trimmings. From n=6 donors, the authors have identified 5 distinct endothelial cell (EC) populations. The expression levels of a set of genes are different among the different donors and different EC clusters. Furthermore, treatment with IL1B, TGFB, or ERGsi decreased the proportion of some of these clusters and increased others, with some migratory and ECM-producing capacity. Another interesting observation in this study is that IL-1B alone induces a shift in the clusters and that is different from the TGFB-induced cells. However ex vivo analyses showed most of the TGFB-induced population are the ones that matched the in vitro observations. Another interesting finding of the work is that the authors detected SNPs linked to chromatin accessibility to the set of genes identified within these EC populations. Overall, the work is intriguing and has some novel aspects to it, especially the link between EC-derived EndMT in culture and comparing that with ex vivo atherosclerotic samples. However, the experiments are lacking in controls, the purity of the isolation, and the use of multiple donors (deceased hearts) to draw conclusions. The lack of validations for the work is a huge concern. Additional major and minor concerns are:Major concerns:(1) Abstract: line 15: ECs are a major cell type in atherosclerosis progression - That is a bold statement: What about macrophages and VSMCs?

We have made changes to this sentence so that it now reads: “Endothelial cells (ECs), macrophages, and vascular smooth muscle cells (VSMCs) are major cell types in atherosclerosis progression, and heterogeneity in EC sub-phenotypes are becoming increasingly appreciated.”

(2) Methods: The cells were isolated from the deceased heart by a device? What kind of device? Is it a standard method, showing a figure or data suggesting the purity of the isolates. Also, the authors mentioned that they assessed EC function, but no single figure suggests that. Why were the cells treated with fibronectin?

We thank Reviewer #1 for bringing this to our attention. We did not isolate and identify the cells ourselves. This was done in a prior study as described in reference 41. The only function of the device was to hold the aortic explanted tissue in place so the luminal surface of the ECs could be digested with collagenase. We have made edits to clarify these points in Methods, “Tissue Procurement and Cell Culture”: “HAECs were isolated from the luminal surface of the aortic trimmings using collagenase, and identified by Navab et al. using their typical cobblestone morphology, presence of Factor VIII-related antigen, and uptake of acetylated LDL labeled with 1,1’-dioctadecyl-1-3,3,3’,3’-tetramethyl-indo-carbocyan-ine perchlorate (Di-acyetl-LD) (42).”

(3) Why did the authors elect to treat each donor cell with different treatment types and different concentrations, also why 1ng/ml of IL-1B?

We have addressed the study design asymmetry above. We chose the treatments because we questioned whether HAECs responded heterogeneously to these stimuli. We were interested in using these stimuli, because they have previously been used in vitro to induce EndMT and/or inflammation, two major pathophysiological processes in CAD. This is outlined in the Introduction: “We also quantified single cell responses to three perturbations known to be important in EC biology and atherosclerosis. The first was activation of transforming growth factor beta (TGFB) signaling, which is a hallmark of phenotypic transition and a regulator of EC heterogeneity (20, 30). The second was stimulation with the pro-inflammatory cytokine interleukin-1 beta (IL1B), which has been shown to model inflammation and EndMT in vitro (31-35), and whose inhibition reduced adverse cardiovascular events in a large clinical trial (36). The third perturbation utilized in our study was knock-down of the ETS related gene (ERG), which encodes a transcription factor of critical importance for EC fate specification and homeostasis (37-41).”

(4) The justification for comparing the EC population in ERGsi is unclear? This was detected as the highest in EC2 but EC2 is not the main cell type across the donors.

We include a justification for comparing the EC populations with siERG in the Introduction:

“There are notable benefits and limitations for studying heterogeneity using in vitro and in vivo approaches in atherosclerosis research. In vitro approaches provide unique opportunities for interrogating consequences of genetic and chemical perturbations in highly controlled environments and are adept at identifying mechanistic relationships on accelerated timelines.”

…and…

“We… quantified single cell responses to three perturbations known to be important in EC biology and atherosclerosis…The third perturbation utilized in our study was knock-down of the ETS related gene (ERG), which encodes a transcription factor of critical importance for EC fate specification and homeostasis (37-41).”

Notably, we found the highest proportion of cells in EC3 with siERG, not EC2:

The one cluster exhibiting increased proportions of cells upon EndMT perturbations was EC3, with 3 of 4 EC IL1B-exposed donors having increased proportions in EC3 (p = 0.08 by 2-sided paired t-test; Figure 3A), 4 of 5 TGFB2-exposed donors having increased proportions (p = 0.04 by 2-sided paired t-test; Figure 3A), and 3 of 3 donors having increased EC3 proportions upon ERG knock-down (Figure 3B).

(5) The different proportions of clusters per donor and their responses are different. These donors are from deceased hearts, could the postmortem induce changes in the ECs? The presence of SMC pathways in their analysis may indicate SMC contamination within the isolation rather than EndMT?

We have now included the possibility of postmortem effects in the Discussion:

“We cannot exclude the possibility that EC3 is an EndMT cluster, although we would have expected more significant deviation from clusters EC1 and EC2. It is also possible that the postmortem could induce changes in the ECs, or that the duration and doses of perturbations chosen were not sufficient to elicit complete EndMT.”

As aforementioned, we addressed the purity of isolation within the Discussion.

(6) Figure 4A is confusing, what do the dots indicate and the intersection size mean? What is the difference between Figure 4 C and 4 E?

We have added a description of rows and columns to the legend for Figure 4A:

“(A), Upset plots of up- and down-regulated DEGs across EC subtypes with siERG (grey), IL1B (pink), and TGFB2 (blue). Upset plots visualize intersections between sets in a matrix, where the columns of the matrix correspond to the sets, and the rows correspond to the intersections. Intersection size represents the number of genes at each intersection.”

Figure 4E depicts up- and down-regulated DEGs that are mutually exclusive and shared between IL1B and siERG in EC3, whereas Figure 4C depecits up- and down-regulated DEGs with IL1B alone compared to siSCR in EC2, EC3, and EC4. This is described within the legend for Figure 4C and Figure 4E:

“(C), PEA for EC2-4 up- and down-regulated DEGs with IL1B compared to control media… (E), PEA comparing up- and down-regulated DEGs that are mutually exclusive and shared between IL1B and siERG in EC3.”

(7) VSMCS 5 in Figure 5 is interesting, but it could be contaminated with SMCs in your EC population and they are SMCs indeed with some mesenchymal transdifferentiation?

As abovementioned, we addressed the purity of isolation within the Discussion.

Minor concerns:(1) All growth supplements, kits, and reagents should be provided with their sources and catalogue numbers.

Sources and catalogue numbers have now been added to the following Methods sections:

“Tissue Procurement and Cell Culture”: “Cells were grown in culture in M-199 (ThermoFisher Scientific, Waltham, MA, MT-10-060-CV) supplemented with 1.2% sodium pyruvate(ThermoFisher Scientific, cat. no. 11360070), 1% 100X Pen Strep Glutamine (ThermoFisherScientific, cat. no. 10378016), 20% fetal bovine serum (FBS, GE Healthcare, Hyclone, Pittsburgh, PA), 1.6% Endothelial Cell Growth Serum (Corning, Corning, NY, cat. no. 356006), 1.6% heparin, and 10µL/50 mL Amphotericin B (ThermoFisher Scientific, cat. no. 15290018). HAECs at low passage (passage 3-6) were treated prior to harvest every 2 days for 7 days with either 10 ng/mL TGFB2 (ThermoFisher Scientific, cat. no. 302B2002CF), IL1B (ThermoFisher Scientific, cat. no. 201LB005CF), or no additional protein, or two doses of small interfering RNA for ERG locus (siERG; Table S18 in the Data Supplement), or randomized siRNA (siSCR; Table S18 in the Data Supplement).”

…and…

“siRNA Knock-down, qPCR, and Western Blotting”: “Knockdown of ERG was performed as previously described (40) using 1 nM siRNA oligonucleotides in OptiMEM (ThermoFisher Scientific, cat. no. 11058021) with Lipofectamine 2000 (ThermoFisher Scientific, cat. no. 11668030).”

(2) The quantification of western blot how?

Methods, “siRNA Knock-down, qPCR, and Western Blotting” now reads: “Western blots were quantified using ImageJ (76).”

(3) All the supplemental figures are listed incorrectly in the manuscript. For example, the authors refer to Figure S11B which should be S10. Please review the manuscript throughout to refer to the correct figures.

We thank Reviewer #1 for bringing this to our attention. Figure S4 was missing, leading to incorrectly listed supplemental figures for Figures S4-S12. Figure S4 has now been included, and Figures S4-S12 are now listed correctly within the manuscript text.

(4) Please refer to IL-1B as IL-1beta, same with TGFB.

We have left the terms as is, since it is also routine to refer to IL-1beta as IL1B, and TGFbeta as TGFB.

(5) here are typos throughout the manuscript, such as 4C, VW Fexpression, VWFand VCAM-1.

We could not locate typos “VW Fexpression” or “VWFand VCAM-1”. We do not consider “4C” a typo, as it refers to the temperature at which the centrifuge was set to in Methods, “Nuclear Dissociation and Library Preparation”: “Samples were centrifuged at 500 rcf for 5 minutes at4C…”

(6) Please define the abbreviations: line 69 and also cite the source of the use of aSMA/PECAM1 as EndMT?

We have now included abbreviation definitions and the cited source for ECs that co-express aSMA/PECAM-1 in atherosclerotic lesions within the Introduction: “These studies have described an unexpectedly large number of cells co-expressing pairs of endothelial and mesenchymal proteins, including fibroblast activating protein/von Willebrand factor (FAP/VWF), fibroblastspecific protein-1/VWF (FSP-1/VWF), FAP/platelet-endothelial cell adhesion molecule-1 (CD31 or PECAM-1), FSP-1/CD31 (20), phosphorylation of TGFB signaling intermediary SMAD2/FGF receptor 1 (p-SMAD2/FGFR1) (22), and α-smooth muscle actin (αSMA)/PECAM-1 (23).”

(7) The changes in % cells in cluster per donor per condition in Figure 3 are interesting, have the authors observed a change of one cluster at the expense of another i.e. do they transdifferentiate into another with different treatments?

Figure 3 shows that as percent of cells in EC3 go up with TGFB or IL1B, they go down in EC4 with these treatments. This has been added to the Discussion: “Moreover, as the percent of cells in EC3 go up with TGFB or IL1B, they go down in EC4, suggesting trans-differentiation from EC4 into EC3 with these perturbations.”

(8) Functional analysis of these clusters with and without treatment is required to confirm the EndMT.

We do not claim that the cells underwent EndMT. Rather, we use pro-EndMT perturbations previously described in the literature to test whether ECs respond heterogeneously to stimuli which are relevant to CAD. We found that EC subtype was a greater determinant of cell state than treatment.

(9) No blank line at 266. The break is in the middle of the sentence, also cytoplasmic cytoplasmic ribosomal proteins (typo?).

We have revised these sentences to read: “Shared IL1B- and siERG-upregulated genes were enriched in COVID-19 adverse outcome pathway (WP4891; p-value 1.9x10-9) (52). Shared IL1B- and siERG-attenuated genes are enriched in several processes involving ribosomal proteins, including ribosome, cytoplasmic (CORUM:306; p-value 3.3x10-7), cytoplasmic ribosomal proteins (WP477; p-value 5.3x10-7), and peptide chain elongation (R-HSA-156902; pvalue 5.9x10-7) (Figure 4E).”

(10) The sentence in line 321 "These observations support ....of human, seems incomplete.

We revised these sentences to read: “Expected pathway enrichments are observed for annotated cell types, including NABA CORE MATRISOME (M5884; p-value 4.8x10-41) for fibroblasts, blood vessel development (GO:0001568; p-value 5.6x10-33) for ECs, and actin cytoskeleton organization (GO:0030036; p-value 1.3x10-15) for VSMCs (Figure S5D-G in the Data Supplement). These observations support the diverse composition of human atherosclerotic lesions.”

(11) What do the authors mean by (at least partially) line 444?

We revised this sentence to read: “In fact, the limited correlation with ex vivo data supports this interpretation.”

(12) Some unrelated data in the paper, like supplemental figure 10B and supplemental figure 11?

These data are relevant to methods, and have been kept.

**Reviewer #2 (Recommendations For The Authors):**
We need this work to expand our knowledge of endothelial biology. Please address my concerns to further strengthen this work.